# The isotopic composition of atmospheric nitrous oxide observed at the high-altitude research station Jungfraujoch, Switzerland

Longfei Yu[1*], Eliza Harris[1†], Stephan Henne[1], Sarah Eggleston[1], Martin Steinbacher[1], Lukas Emmenegger[1], Christoph Zellweger[1] and Joachim Mohn[1]

[1]Laboratory for Air Pollution & Environmental Technology, Empa, Swiss Federal Laboratories for Materials Science and Technology, Ueberlandstr. 129, CH-8600 Duebendorf, Switzerland.

[†]Current address: Institute of Ecology, University of Innsbruck, Sternwartestrasse 15, A-6020 Innsbruck, Austria

[*] Correspondence: L. Yu (longfei.yu@empa.ch)

## Abstract

Atmospheric nitrous oxide ($N_2O$) levels have been continuously growing since preindustrial times.
Mitigation requires information about sources and sinks on the regional and global scales. Isotopic
composition of $N_2O$ in the atmosphere could contribute valuable constraints. However, isotopic
records of $N_2O$ in the unpolluted atmosphere remain too scarce for large-scale $N_2O$ models. Here,
we report the results of discrete air samples collected weekly to bi-weekly over a five-year period
at the high-altitude research station Jungfraujoch, located in central Switzerland. High-precision
$N_2O$ isotopic measurements were made using a recently developed preconcentration-laser
spectroscopy technique. The measurements of discrete samples were accompanied by *in situ*
continuous measurements of $N_2O$ mixing ratios. Our results indicate a pronounced seasonal pattern
with minimum $N_2O$ mixing ratios in late summer, associated with a maximum in $\delta^{15}N^{bulk}$ and a
minimum in intramolecular $^{15}N$ site preference ($\delta^{15}N^{SP}$). This pattern is most likely due to
stratosphere-troposphere exchange (STE), which delivers $N_2O$-depleted but $^{15}N$-enriched air from
the stratosphere into the troposphere. Variability in $\delta^{15}N^{SP}$ induced by changes in STE may be
masked by biogeochemical $N_2O$ production processes in late summer, which are possibly
dominated by a low-$\delta^{15}N^{SP}$ pathway of $N_2O$ production (denitrification), providing an explanation
for the observed seasonality of $\delta^{15}N^{SP}$. Footprint analyses and atmospheric transport simulations
of $N_2O$ for Jungfraujoch suggest that regional emissions from the planetary boundary layer
contribute to seasonal variations of atmospheric $N_2O$ isotopic composition at Jungfraujoch, albeit
more clearly for $\delta^{15}N^{SP}$ and $\delta^{18}O$ than for $\delta^{15}N^{bulk}$. With the time-series of five years, we obtained
a significant interannual trend for $\delta^{15}N^{bulk}$ after deseasonalization (-0.052±0.012‰ $a^{-1}$), indicating
that the atmospheric $N_2O$ increase is due to isotopically depleted $N_2O$ sources. We estimated the
average isotopic signature of anthropogenic $N_2O$ sources with a two-box model to be -8.6±0.6‰
for $\delta^{15}N^{bulk}$, 34.8±3‰ for $\delta^{18}O$ and 10.7±4‰ for $\delta^{15}N^{SP}$. Our study demonstrates that seasonal
variation of $N_2O$ isotopic composition in the background atmosphere is important when
determining interannual trends. More frequent, high-precision and inter-laboratory compatible
measurements of atmospheric $N_2O$ isotopocules, especially for $\delta^{15}N^{SP}$, are needed to better
constrain anthropogenic $N_2O$ sources, and thus the contribution of biogeochemical processes to
$N_2O$ growth on the global scale.

# 1 Introduction

Nitrous oxide ($N_2O$) is a potent greenhouse gas (Fowler et al., 2015) and a strong stratospheric ozone-depleting substance (Ravishankara et al., 2009). For several decades, near-surface atmospheric $N_2O$ mixing ratios have been continuously measured at a series of remote sites, within the networks of the Global Atmosphere Watch Programme (JMA and WMO, 2018), the Advanced Global Atmospheric Gases Experiment (AGAGE) (Prinn et al., 2018), and the National Oceanic and Atmospheric Administration (NOAA) Earth System Research Laboratory (ESRL) Global Monitoring Division (GMD) (Nevison et al., 2011). These measurements have shown a significant increase in atmospheric $N_2O$ mixing ratio, at a current growth rate of about 0.93 nmol mol$^{-1}$ a$^{-1}$ (WMO, 2018). On the global scale, given excessive nitrogen (N) fertilizer application, agriculture is known to be the largest and most important anthropogenic source of $N_2O$ (Reay et al., 2012; Tian et al., 2019). However, long-term observations of $N_2O$ in the unpolluted atmosphere have shown seasonal and interannual variabilities as well as interhemispheric differences in $N_2O$ mixing ratios (Nevison et al., 2011; Thompson et al., 2014a, 2014b), which cannot yet be resolved by atmospheric transport models and existing emission inventories. Moreover, regional contributions of $N_2O$ emissions and the strengths of individual $N_2O$ production pathways remain difficult to quantify.

Isotopic signatures of atmospheric $N_2O$ can provide important constraints on $N_2O$ sources (Denk et al., 2017) and trends (Kim and Craig, 1993). The ratios of $^{15}N/^{14}N$ and $^{18}O/^{16}O$ in $N_2O$ are often reported in $\delta$ notation as $\delta(^{15}N/^{14}N)$ and $\delta(^{18}O/^{16}O)$, abbreviated as $\delta^{15}N^{bulk}$ (average for $^{14}N^{15}N^{16}O$ and $^{15}N^{14}N^{16}O$) and $\delta^{18}O$, respectively. A large fraction of $N_2O$ emitted to the atmosphere originates from soil bacterial processes, which usually emit $N_2O$ that is more enriched in light ($^{14}N$, $^{16}O$) isotopes than the tropospheric background (Pérez et al., 2001; Snider et al., 2015a; Toyoda et

al., 2017). By contrast, $N_2O$ produced in the oceans (Bourbonnais et al., 2017; Fujii et al., 2013)
and emitted from fossil fuel combustion (Ogawa and Yoshida, 2005; Toyoda et al., 2008) has
higher $\delta^{15}N^{bulk}$ and $\delta^{18}O$ values which are comparable to the tropospheric background. A recent
study has summarized isotopic signatures of anthropogenic $N_2O$ sources divided into the EDGAR
(Emissions Database for Global Atmospheric Research) emission categories (Janssens-Maenhout
et al., 2019), showing differences in isotopic signatures between agricultural ($\delta^{15}N^{bulk}$ = -17.8 to -
1.0‰ and $\delta^{18}O$ = 23.9 to 29‰) and industrial sources ($\delta^{15}N^{bulk}$ = -28.7 to 5.5‰ and $\delta^{18}O$ = 28.6
to 40.3‰) (Harris et al., 2017). These empirical ranges, together with isotopic mixing models,
provide a valuable approach to interpret variability in atmospheric $N_2O$ mixing ratios.
A number of studies have analyzed temporal trends in $N_2O$ isotopic composition in the modern
atmosphere (Kaiser et al., 2003; Park et al., 2012; Röckmann and Levin, 2005; Toyoda et al., 2013)
and in the past from firn and ice cores (Bernard et al., 2006; Ishijima et al., 2007; Prokopiou et al.,
2018; Röckmann et al., 2003; Sowers et al., 2002). These isotopic measurements have shown a
decrease in both $\delta^{15}N^{bulk}$- and $\delta^{18}O$-$N_2O$ associated with an increasing trend in atmospheric $N_2O$
mixing ratios since preindustrial times, indicating that the recent increase of atmospheric $N_2O$ may
be due to agricultural emissions ($^{15}N$ and $^{18}O$ depleted). The reported trend since the 1960s seems
rather steady (-0.034±0.005 ‰ $a^{-1}$ for $\delta^{15}N^{bulk}$ and -0.016 ‰±0.006 $a^{-1}$ for $\delta^{18}O$) (Bernard et al.,
2006; Ishijima et al., 2007; Park et al., 2012; Prokopiou et al., 2017; Röckmann et al., 2003;
Röckmann and Levin, 2005). However, a more recent (1999-2010) study reported a smaller
decreasing trend in $\delta^{15}N^{bulk}$ and only an insignificant trend in $\delta^{18}O$ for the Northern Hemisphere
(Toyoda et al., 2013). Several hypotheses were proposed to explain the differences in the observed
trends: 1) the interhemispheric difference in $N_2O$ emission sources results in inconsistent isotopic
signatures among different studies (Thompson et al., 2014b); 2) uncertainties in isotopic
measurements and variable sampling schemes (air type, sampling frequency and time) mask the
small secular trend of $N_2O$ isotopic composition in the background atmosphere (Toyoda et al.,
2013); and/or 3) $N_2O$ source isotopic signatures have changed in recent years, possibly due to
shifts in N fertilizer type and climatic forcing (Tian et al., 2019). Hence, further investigation into
the global $N_2O$ source inventory and its evolution over time requires more frequent, precise
measurements of $N_2O$ isotopocules in the unpolluted atmosphere, particularly in the Northern
Hemisphere.
Recently, site-specific composition of $N_2O$ isotopomers (site preference: $\delta^{15}N^{SP}$), which denotes
the difference of $^{15}N$ between the central ($^{14}N^{15}N^{16}O$, $\alpha$ position) and terminal ($^{15}N^{14}N^{16}O$, $\beta$
position) N atoms, has been applied to constrain sources contributing to atmospheric $N_2O$ (Toyoda
et al., 2013; Yoshida and Toyoda, 2000). $\delta^{15}N^{SP}$ of $N_2O$ is particularly effective for distinguishing
between the major $N_2O$ production processes, i.e. nitrification and denitrification, generally
referred to as aerobic and anaerobic $N_2O$ production, with high and low $\delta^{15}N^{SP}$, respectively (Sutka
et al., 2006). However, despite the advantages of $\delta^{15}N^{SP}$ measurements, existing long-term studies
have not yet been able to reach a definitive understanding of the $\delta^{15}N^{SP}$-$N_2O$ trend, showing both
positive (Bernard et al., 2006; Park et al., 2012; Röckmann and Levin, 2005) and negative
tendencies (Röckmann et al., 2003) over the last four decades. This is probably due to an
insufficient analytical precision and poor inter-laboratory agreement, in particular as the
aforementioned studies are all based on isotope ratio mass spectrometry (IRMS). To retrieve site-
specific isotopic information by IRMS, the $N_2O^+$ molecular ions and the $NO^+$ fragment ions are
analyzed and raw data have to be corrected for rearrangements of central and terminal N and $^{17}O$
content (Toyoda et al., 2001). Inappropriate correction algorithms and the limited availability of
reference materials (Ostrom et al., 2018) further enlarge the analytical uncertainty (Mohn et al.,

109     2014).

Seasonal variability in atmospheric $N_2O$ isotopic composition, which could affect the longer-term
trends, is still rarely reported in the literature (Park et al., 2012; Toyoda et al., 2013). Moreover,
studies of seasonality of $N_2O$ isotopic composition are limited to the recent past since the air
samples derived from firn and ice cores suffer from coarse temporal resolution (< 2 samples per
year). Park et al. (2012) studied seasonality of atmospheric $N_2O$ isotopic composition by analyzing
a set of archived air samples collected from Cape Grim (Australia) using a sophisticated
mathematical modeling approach. They found consistent seasonal patterns in $\delta^{15}N^{bulk}$, $\delta^{18}O$ and
$\delta^{15}N^{SP}$ of atmospheric $N_2O$, showing highest $^{15}N/^{18}O$ enrichment in June and lowest in December.
This pattern was negatively correlated with the seasonality of the $N_2O$ mixing ratios (lowest in
April-May and highest in December), which is in agreement with a previous study by Nevison et
al. (2011). The negative correlation between isotopic composition and mixing ratios has been
explained by stratosphere-troposphere exchange (STE), which transports $N_2O$-depleted but
isotopically enriched stratospheric air (prevailing reduction process) into the lower atmosphere
(Yung and Miller, 1997). However, in a more recent study from Hateruma Island (Japan), Toyoda
et al. (2013) reported insignificant seasonal patterns in atmospheric $N_2O$ isotopocules (smaller
variability than measurement precision), despite their finding of a somewhat similar seasonal
pattern in $N_2O$ mixing ratio (minimum in July). Although there are interhemispheric differences
in $N_2O$ sources and distinct sampling frequencies in the two studies discussed above (2-3 times
per year versus monthly), it is noteworthy that both studies observed significantly larger variability
in $\delta^{15}N^{SP}$ than in $\delta^{15}N^{bulk}$ and $\delta^{18}O$. Whether the fluctuations in $\delta^{15}N^{SP}$ are mainly caused by the
limited repeatability of the chosen analytical techniques or interplay of processes or mechanisms
regulating atmospheric $N_2O$ remains to be tested (Park et al., 2012).
With inherent selectiveness, in particular for site-specific isotopic composition, laser spectroscopy
provides a new analytical approach for direct, precise measurements of all four $N_2O$ isotopocules
(Harris et al., 2014; Mohn et al., 2012). The recent development of quantum cascade laser
absorption spectroscopy (QCLAS) coupled with an automated preconcentration unit has been
applied to measure $N_2O$ isotopocules in ambient air, with comparable precision for $\delta^{15}N^{bulk}$ and
$\delta^{18}O$ and superior precision for $\delta^{15}N^{SP}$ relative to IRMS systems (Harris et al., 2017; Mohn et al.,
2014). Here, we present results from the application of a preconcentration unit coupled to QCLAS
to measure atmospheric $N_2O$ isotopocules in background air collected at the high altitude research
station Jungfraujoch, Switzerland. Between April 2014 and December 2018, we collected weekly
to bi-weekly air samples for $N_2O$ isotopic analyses, in parallel with online measurement of $N_2O$
mixing ratios. To our knowledge, this work reports the first time-series of background atmospheric
$N_2O$ isotopic composition using laser spectroscopy. With this unique dataset, we aim to 1)
constrain seasonal patterns of three $N_2O$ isotopic signatures at the Jungfraujoch observatory; 2)
determine interannual trends in $N_2O$ isotopocules, especially $\delta^{15}N^{SP}$; and 3) interpret the observed
patterns in $N_2O$ mixing ratios using temporal trends in $N_2O$ isotopic composition and reported
isotopic signatures of anthropogenic sources.

## 2 Materials and Method

### 2.1 Site description

The high altitude research station Jungfraujoch (3580 m above sea level), located on the northern ridge of the Swiss Alps, is a well-established site for studying unpolluted atmosphere over Central Europe (e.g. Buchmann et al., 2016). Although the station is located in the free troposphere most of the time, it is occasionally affected by air recently lifted from the planetary boundary layer (Herrmann et al., 2015; Zellweger et al., 2003). Henne et al. (2010) investigated the representativeness of 35 European monitoring stations and categorized Jungfraujoch as "mostly remote". The Jungfraujoch station is part of several national and international networks, like the meteorological SwissMetNet network operated by MeteoSwiss, the Swiss National Air Pollution Monitoring Network (NABEL), the Global Atmospheric Watch Programme (GAW) of the World Meteorological Organization (WMO) and the Integrated Carbon Observation Systems (ICOS) Research Infrastructure. This results in an extended set of long-term and continuously available parameters such as meteorological variables (Appenzeller et al., 2008), greenhouse gases (Schibig et al., 2015; Sepúlveda et al., 2014; Yuan et al., 2018), $CO_2$ isotopic composition (Sturm et al., 2013; Tuzson et al., 2011), ozone-depleting substances and their replacement products (Reimann et al., 2008), atmospheric pollutants (Logan et al., 2012; Pandey Deolal et al., 2012; Zellweger et al., 2009) and aerosol parameters (Bukowiecki et al., 2016).

### 2.2 *In situ* measurements and discrete air sampling (flasks)

*In situ* observations of $N_2O$ mixing ratios commenced at Jungfraujoch in December 2004. Initially, measurements were made with gas chromatograph (GC) (Agilent 6890N, USA) followed by electron capture detection (ECD). The time resolution of these measurements was 24 to 30 minutes.

In late 2014, we implemented a cavity-enhanced off-axis integrated cavity out-put spectroscopy
analyzer (OA-ICOS, Los Gatos Research Inc., Mountain View, CA, USA), which measures the
atmospheric $N_2O$ mixing ratio continuously. Measurements of $N_2O$ mixing ratios at Jungfraujoch
were calibrated with three standard gases (319, 327 and 342 nmol mol$^{-1}$) and accompanied with
measurement of a working standard (331 nmol mol$^{-1}$) every 160 minutes to account for
instrumental drift. In addition, short- (two times every 40 hours) and long-term (every 40 hours)
target measurements were included to monitor the data quality. Due to the superior measurement
precision compared to the GC-ECD method (Lebegue et al., 2016), the OA-ICOS record has
become the primary time-series since January 2015. The GC-ECD observations continued until
summer 2016 for comparison and quality control.
Additional parameters, recorded within the NABEL network and the ICOS infrastructure, were
included in the analysis below. These data were carbon monoxide (CO) (measured by cavity ring-
down spectroscopy; Model G2401, Picarro Inc., USA), the sum of oxidized nitrogen species ($NO_y$)
(measured by chemiluminescence detection after conversion of $NO_y$ to NO on a heated gold
catalyst; CLD 89p, Eco Physics, Switzerland) and $O_3$ (measured by UV absorption; TEI 49i,
Thermo Scientific, USA). Details on measurement methods and calibration strategies can be found
in Zellweger et al. (2009) for CO, Pandey Deolal et al. (2012) for $NO_y$ and Logan et al. (2012) for
$O_3$.
In conjunction with the online measurements, we deployed an automated sampling system (Fig.
S1) to collect pressurized air samples in aluminum cylinders from the same air inlet at the Sphinx
observatory of the Jungfraujoch station, for subsequent $N_2O$ mixing ratio and isotopic analyses.
The sample collection was conducted weekly from April 2014 to February 2016. After a sampling
gap of five months due to a technical failure, we reinitiated a bi-weekly sampling, which continued
from August 2016 to December 2018. The sampling system, automated by a customized LabVIEW
program (National Instruments Corp., USA), consisted of a Nafion drier (PD-100T-48MSS, Perma
Pure LLC, USA), a membrane gas compressor (KNF Neuberger, USA; Type N286 series), a 16-
port selector valve (EMT2CSD16MWEPH, VICI AG, Swtizerland), and a rack to accommodate
nine 2-L aluminum flasks (Luxfer, Messer Schweiz AG, Switzerland). During sample filling, pre-
evacuated flasks were first purged with ambient air five times (1 hour), and then filled to 12000
hPa within 40 min, resulting in approximately 24 L (298 K and 1000 hPa) of air per flask for
isotopic analysis. Air sample filling generally took place between 2:00 and 3:00 pm local time at
each sampling day. Sample flasks were sent back to the laboratory at Empa for analyses every few
months. For this study, 142 air samples were collected in flasks and analyzed for $N_2O$ isotopocules.
**2.3 Analyses of discrete air samples**
Discrete air samples were regularly analyzed in batches but note in chronological order to prevent
the imprint of analytical drifts on temporal trends of the samples. $N_2O$ mole fractions were
analyzed by QCLAS (CW-QC-TILDAS-76-CS, Aerodyne Research Inc., USA) against NOAA
standards on the WMO-X2006A calibration scale (Hall et al., 2007), at a precision around 0.1
nmol mol$^{-1}$ (determined with the average of 1-min data).
The four most abundant $N_2O$ isotopocules ($^{14}N^{14}N^{16}O$, 99.03%; $^{14}N^{15}N^{16}O$, 0.36%; $^{15}N^{14}N^{16}O$,
0.36%; $^{14}N^{14}N^{18}O$, 0.20%) were analyzed using a customized QCLAS system (Aerodyne Research,
Inc., USA) (Heil et al., 2014) coupled with an automated preconcentration device (Mohn et al.,
2010). Before entering the pre-concentration unit, sample air is passed through a Sofnocat 423 trap
(Molecular Products Limited, GB) to remove CO, and subsequently through an Ascarite trap
(Ascarite: 6 g, 10–35 mesh, Sigma Aldrich, Switzerland, bracketed by $Mg(ClO_4)_2$, 2 × 1.5 g, Alfa
Aesar, Germany) to remove $CO_2$ and water. Approximately 5.5 L of air with a flow of 250 ml min$^-$
$^1$ (at 295 K and 3500 hPa) is then passed through a HayeSep D trap cooled to -145 °C to collect
N$_2$O (Mohn et al., 2010). For N$_2$O release to the multipath cell of the QCLAS, the HayeSep D trap
is quickly heated to 10 °C and flushed with high-purity synthetic air (20.5% of O$_2$ in N$_2$) carrier
gas at a flow rate of 25 ml min$^{-1}$ (at 295 K and 3500 hPa). A final cell pressure around 16 hPa is
achieved, which results in an N$_2$O mixing ratio of about 45 μmol mol$^{-1}$. More instrumental details
can be found in previous studies (Harris et al., 2017; Mohn et al., 2010, 2012). Sample tanks were
each analyzed twice to yield duplicates for N$_2$O isotopic results, which left sufficient air for amount
fraction analysis as described in the previous paragraph.
**2.4 Data analyses**
We used 10-minute averages of the continuous *in situ* measurements from the Jungfraujoch station
across this study. For a point-to-point comparison of continuous and discrete measurements of
N$_2$O mixing ratio, we aggregated 10-minute averages of *in situ* data for the same period when the
discrete sample was filled into the cylinder (40 min).
In this study, we report abundances of N$_2$O isotopocules using $\delta$ notation (‰) as below:

$$\delta X = \frac{(R_{sample} - R_{standard})}{R_{standard}} \qquad (1)$$

where X refers to $^{15}N^{\alpha}$ ($^{14}N^{15}N^{16}O$), $^{15}N^{\beta}$ ($^{15}N^{14}N^{16}O$) and $^{18}O$ ($^{14}N^{14}N^{18}O$); R refers to the ratio
between the amount fractions of the rare isotopocules as mentioned above and the amount fraction
of $^{14}N^{14}N^{16}O$; isotope standards refer to atmospheric N$_2$ for $^{15}N$ and Vienna Standard Mean Ocean
Water (VSMOW) for $^{18}O$.
Hence, the total $^{15}N$ content of N$_2$O and site-specific composition of N$_2$O isotopomers could be
further illustrated as $\delta^{15}N^{bulk}$ and $\delta^{15}N^{SP}$, respectively, according to the equations below:

$$\delta^{15}N^{bulk} = (\delta^{15}N^{\alpha} + \delta^{15}N^{\beta})/2 \qquad (2)$$

$$\delta^{15}N^{SP} = \delta^{15}N^{\alpha} - \delta^{15}N^{\beta} \qquad (3)$$

Two standards (CG1 and CG2; in 79.5% $N_2$ and 20.5% $O_2$) with distinct isotopic signatures ($\delta^{15}N^{\alpha}$ = 16.29 ± 0.07‰ (CG1) and -51.09 ± 0.07‰ (CG2); $\delta^{15}N^{\beta}$ = -2.59 ± 0.06‰ and -48.12 ± 0.04‰; $\delta^{18}O$ = 39.37 ± 0.04‰ and 30.81 ± 0.03‰) were used for calibrating isotopic composition. The calibration gases CG1 and CG2 were calibrated on the Tokyo Institute of Technology (TIT) scale, based on cross-calibration with primary standards assigned by TIT (Mohn et al., 2012, 2014). In addition, CG1 was measured repeatedly between samples and target gases to account for instrumental drift. Both CG1 and CG2 have $N_2O$ mixing ratios of 45 μmol mol$^{-1}$, similar to the $N_2O$ amount fraction of the samples after preconcentration. However, to correct for possible instrumental dependence on $N_2O$ mixing ratio, CG1 was diluted to $N_2O$ mole fractions of 35-40 μmol mol$^{-1}$ within each measurement batch. In general, duplicated isotopic measurements of flask samples yielded values of repeatability of 0.10-0.20‰ for $\delta^{15}N^{bulk}$ and $\delta^{18}O$, and 0.15-0.25‰ for $\delta^{15}N^{SP}$.

At the beginning of the project, a batch of three cylinders (50 L water volume, Luxfer, Italy) were filled with pressurized ambient air in Dübendorf with an oil-free, three stage compressor (SA-3, Rix Industries, USA) and used as long-term target gases. The pressurized ambient air target gas was analyzed with identical treatment as Jungfraujoch air samples during every analysis batch, to monitor long-term analytical drift. Standard deviations for repeated target gas measurements throughout the period of Jungfraujoch sample measurements, were 0.13‰ for $\delta^{15}N^{bulk}$, 0.21‰ for $\delta^{15}N^{SP}$, and 0.11‰ for $\delta^{18}O$ (Fig. S2).

**2.5 Surface air footprint analysis and simulated regional $N_2O$ enhancement**

We analyzed the air mass origin at Jungfraujoch by applying the Lagrangian particle dispersion
model (LPDM) FLEX-PART in the backward mode (Stohl et al., 2005). The model was driven by
meteorological fields taken from the ECMWF-IFS operational analysis cycle, extracted at a
resolution of $1°\times1°$, 90/137 levels globally, and at higher horizontal resolution of $0.2°\times0.2°$ for
central Europe. We released 50000 virtual air parcels every 3 hours at 3000 m a.s.l. from
Jungfraujoch to perform backward dispersion simulations over 10 days, which allowed us to
calculate surface source sensitivities (concentration footprints). A release height of 3000 m a.s.l.
was previously determined to be an optimum for simulating concentration footprints at
Jungfraujoch, given the stated horizontal resolution which results in a considerable smoothing of
the complex, alpine orography (Keller et al., 2012). The 3-hourly surface footprints for the whole
observation period were used to categorize different transport regimes using the clustering
approach outlined in Sturm et al. (2013). This allowed us to distinguish among six different source
regions: Free Troposphere (FT), Southwest (SW), East (E), Local (L), West (W) and Northwest
(NW).
Similar to Henne et al. (2016) for $CH_4$ and based on spatially resolved $N_2O$ emission inventories
(Meteotest for Switzerland; EDGAR for Europe), we used the FLEXPART concentration
footprints to calculate time-series of atmospheric mole fraction increases at Jungfraujoch resolved
by emission sectors (Henne et al., 2016). The emission inventory by Meteotest consists of 12
emission sectors, among which all sectors except "organic soils" are comparable to sectors in the
EDGAR inventory (See Table 1) (Janssens-Maenhout et al., 2019). To improve seasonal
representation of the emissions in our model, we used a monthly resolved, optimized version of
the emission inventory, which was obtained through inverse modeling using the $N_2O$ atmospheric
mole fractions observed between March 2017 and September 2018 at the tall tower site
Beromuenster on the Swiss plateau (Henne et al., 2019). Therefore, in this study, source
contributions to Jungfraujoch were estimated specifically for the period mentioned above.
**2.6 Evaluation of seasonal pattern and interannual trend for time-series**
To explore seasonality and interannual trends, we fit the time-series of *in situ* measurements of
$N_2O$ and $O_3$ mixing ratios, $NO_y$-to-CO ratios and isotopic measurements of $N_2O$ with polynomial
functions and Fourier series (four harmonics for *in situ* measurements and two harmonics for
discrete measurements) (Thoning et al., 1989). Time-series were then decomposed into a linear
trend, seasonal variability (per 12 months) and residuals. This fit was conducted with a nonlinear
least-squares (NLS) model with R-3.5.3 (R Core Team, 2016). The detrended seasonality was
examined by comparing peak-to-peak amplitudes with our analytical precisions and the
uncertainty given by the one standard deviation of monthly residuals. To determine interannual
trends, a linear regression was applied to both the raw and the deseasonalized datasets. The
significance level is set to $p < 0.01$. The interannual trends for $N_2O$ mixing ratios were found to be
little affected by seasonality, so growth rates were determined only based on the raw datasets.
Although Jungfraujoch is a remote site, episodic influence from the planetary boundary layer can
be observed at the station (Pandey Deolal et al., 2012; Zellweger et al., 2003). For evaluating trends
of $N_2O$ mixing ratio measurements, we filtered out *in situ* data with significant influence of plenary
boundary layer, in order to represent a major air mass footprint from the free troposphere (FT). In
addition to the air transport regimes, an alternative filtering criterion for the free troposphere was
based on the published mean ranges of $NO_y$ mixing ratios (501-748 pmol mol$^{-1}$ depending on the
season) and $NO_y$ to CO ratios (0.003-0.005 depending on the season) at Jungfraujoch (Zellweger
et al., 2003). This criterion is less strict than that given by footprint analyses (Herrmann et al.,
2015). After applying this criterion to the isotopic time-series (which led to the exclusion of 32
measurement points), we re-evaluated the seasonal and interannual trends in the $N_2O$ isotopic
composition. In addition, because of the strong variability observed for isotopic data during the
first 1.5 years (until February 2016), we performed an independent evaluation for the time-series
starting from August 2016.
**2.7 Two-box model simulation**
A two-box model representing a well-mixed troposphere and stratosphere was used to estimate the
anthropogenic $N_2O$ source strength and isotopic composition from the trends measured at
Jungfraujoch, similar to the approaches used by several previous studies (Ishijima et al., 2007;
Röckmann et al., 2003; Schilt et al., 2014; Sowers et al., 2002). The input variables used to run the
model are given in Table 2. 200 iterations of the model were run using a Monte Carlo-style
approach to approximate the uncertainty considering the uncertainty distribution for each input
variable as given in Table 2. All variables were set independently within the Monte Carlo
approximation except for preindustrial $N_2O$ life time ($\tau_{PI}$), which was fixed to 106% of the present-
day $N_2O$ life time $\tau_{PD}$ (Prather et al., 2015).
Within each iteration of the model, the preindustrial $N_2O$ burden was first described, assuming
steady state in the preindustrial era. The preindustrial stratospheric $N_2O$ mixing ratio ($c_{S,PI}$)
($270\pm7.5$ nmol mol$^{-1}$) was taken from Sowers et al. (2002):

$$0 = F_{ex}\left(c_{PI} - c_{S,PI}\right) - (M_{PI} + M_{S,PI})/\tau_{PI} \tag{4}$$

where $F_{ex}$ refers to the troposphere-stratosphere exchange rate; $c_{PI}$ refers to the preindustrial
tropospheric $N_2O$ mixing ratio; and $M_{PI}$ and $M_{S,PI}$ are the masses of $N_2O$ in the troposphere and
stratosphere respectively. The preindustrial terrestrial flux in Sowers et al. (2002) (equation 2) was
used here assuming no anthropogenic emissions. The delta values for the preindustrial stratosphere
and the fractionation factor for the stratospheric sink were taken from equations 6 and 7 from
Sowers et al. (2002) assuming steady state and no anthropogenic emissions. The model was run
with a yearly time step starting from the preindustrial assuming that anthropogenic emissions
began in 1845 (Sowers et al., 2002). For each year of the model run, the anthropogenic flux was
calculated according to the exponential increase described by Sowers et al. (2002):
$$F_{\mathrm{anth},t} = e^{\alpha(t-t_0)} - 1 \qquad (5)$$

where $t$ is the current year, $t_0 = 1845$ and $\alpha$ is the growth rate (assumed to be constant). The rates
of change for tropospheric and stratospheric $N_2O$ mixing ratios were then retrieved from equations
2 and 3 in Sowers et al. (2002), and for the isotopic composition of stratospheric and tropospheric
$N_2O$ from equations 6 and 7 in Sowers et al. (2002).
The values of the parameters describing the anthropogenic flux were optimized to fit both the trend
and the absolute values for the five years of Jungfraujoch isotope data, and the mixing ratio data
from the Jungfraujoch flasks and *in situ* data since 2005 (GAW data source). The uncertainties in
$\alpha$ and in the anthropogenic source isotopic signatures were approximated by one standard
deviation of values derived from repeated model runs.
**2.8 Evaluation of the combined effects from STE and soil emission on $\delta^{15}N^{SP}$**
To evaluate the combined effects of STE and soil emission on the seasonal variability of $\delta^{15}N^{SP}$
(i.e. August minima), we made a mixing calculation as below:
*Soil emission*: Based on the determined seasonality of $N_2O$ mole fraction at Jungfraujoch, the
maximum $N_2O$ mole fraction enhancement was approximately 0.2 nmol mol$^{-1}$ above baseline (Fig.
1). Hence, we assumed $N_2O$ enhancement from soil emission to be close to 0.15 to 0.20 nmol mol⁻
¹, which is close to the maximum $N_2O$ enhancement in our observation. The isotopic effect from
soil emission can be derived from the difference between soil emission (7.2‰; Table 1) and
tropospheric air (18‰, Fig. 2) in $\delta^{15}N^{SP}$, i.e. -10.8‰.
*Mixing with stratospheric air*: The minimum of $N_2O$ mole fraction in August (-0.20 nmol mol⁻¹)
is likely to be the result of both $N_2O$ mole fraction enhancement from soil emission and $N_2O$ mole
fraction depletion due to STE. Given the assumed $N_2O$ enhancement from soil emission, we
estimated the $N_2O$ depletion due to STE as -0.35 to -0.40 nmol mol⁻¹. The isotopic effect due to
mixing with stratospheric air can be approximated using the apparent isotopic fractionation $\varepsilon_{app}$
(Kaiser et al., 2006), which was derived from the slope of Rayleigh plot with normalized $N_2O$
mole and isotope ratios. For $^{15}N^{SP}$, $\varepsilon_{app}$ is calculated from the difference between $^{15}N/^{14}N$ isotope
fractionations at the central and terminal N atoms, i.e. $^{\alpha}\varepsilon_{app}$ - $^{\beta}\varepsilon_{app}$. Therefore, for the lower
stratosphere, $\varepsilon_{app}(^{15}N^{SP})$ was calculated to be about -15‰ (see more details in Kaiser et al., 2006).
*Overall effect*: Combing the isotope effects and contributions to the change of $N_2O$ mole fraction
by the two processes, the net effect is [(-0.35 to -0.40 nmol mol⁻¹) (-15‰) + (0.15 to 0.20 nmol
mol⁻¹) ( -10.8‰)] / (330 nmol mol⁻¹) $\approx$ 0.01‰. Such isotope effect is below our analytical
precision and too small to be measured in the background atmosphere.
**2.9 "Bottom-up" estimates of source isotopic signatures**
To gauge the accuracy of the two-box model, we deployed a "bottom-up" approach as an
alternative method of estimating the $N_2O$ source signatures. The isotopic signatures of most $N_2O$
source sectors given in the Meteotest/EDGAR emission inventory are available from the literature,
except for the "Refinery" (Table 1). As "Refinery" generally contributes only about 0.02% of the
N$_2$O emission at Jungfraujoch, it was excluded for source isotopic signature estimation. The
simulated N$_2$O emissions by variable sources were categorized according to the EDGAR emission
types (Janssens-Maenhout et al., 2019). We then calculated isotopic signatures for the overall
source and the anthropogenic sources alone (excluding indirect natural emission) as weighted
averages.

# 3 Results

## 3.1 Atmospheric N$_2$O mixing ratios at Jungfraujoch

We observed a linear growth of atmospheric N$_2$O at Jungfraujoch during the period 2014-2018 (Fig. 1a). A point-to-point comparison of discrete and *in situ* measurements showed good agreement, in particular after the first year (2015-2018), where the data quality of *in situ* measurements was largely improved due to the implementation of the more precise laser spectroscopy method as compared to GC-ECD (Fig. 1b). The improvement in analytical precision for N$_2$O mixing ratio was due to better temporal coverage by the OA-ICOS instrument, in contrast with the GC analyses which conduct one measurement per 24-30 minutes. The annual growth rates from 2014 to 2018 determined with *in situ* measurements were $0.880 \pm 0.001$ and $0.993 \pm 0.001$ nmol mol$^{-1}$ a$^{-1}$ with and without GC-ECD measurements in 2014, respectively. This difference in N$_2$O growth rates is probably due to the limited data quality of GC-ECD, although a lower growth rate in 2014 compared to 2015-2018 cannot be excluded. It is noteworthy that the N$_2$O growth rate determined for 2015 to 2019 at Jungfraujoch is slightly above the global mean growth rate for the recent decade reported by NOAA ($0.93 \pm 0.03$ nmol mol$^{-1}$ a$^{-1}$) (WMO, 2018). If we filter the *in situ* dataset to examine only the "free troposphere" periods, we obtain a lower increase ($0.858 \pm 0.002$ nmol mol$^{-1}$ a$^{-1}$). By comparison, the absolute growth rate determined from the discrete gas samples was even lower albeit larger uncertainty ($0.813 \pm 0.027$ nmol mol$^{-1}$ a$^{-1}$).

A significant seasonal pattern was observed for N$_2$O mixing ratios measured *in situ*, with a maximum in early summer and a minimum in late summer (Fig. 1c). For discrete N$_2$O measurements a similar trend was observed, but the detrended seasonality was not significant (Fig. S3), which might be due to the much lower number of samples.

## 3.2 Interannual trends of N$_2$O isotopic composition and anthropogenic source signatures

Time-series of $\delta^{15}N^{bulk}$, $\delta^{15}N^{SP}$ and $\delta^{18}O$ for atmospheric N$_2$O at Jungfraujoch are shown in Figure 2. The NLS model simulation accounts well for the variabilities of isotopic time-series. Interannual trends of three isotopic deltas were determined for both raw and deseasonalized datasets by linear regression (Table 3). The deseasonalized interannual trends were slightly smaller than the trends determined with the raw datasets. For the whole dataset, the deseasonalized trend indicates a significant decrease in $\delta^{15}N^{bulk}$, of -0.052±0.012‰ a$^{-1}$. In contrast, deseasonalized time-series of $\delta^{15}N^{SP}$ and $\delta^{18}O$ increased, albeit insignificantly, by 0.065±0.027‰ a$^{-1}$ and 0.019±0.011‰ a$^{-1}$, respectively. The trends determined for periods with major air mass footprints from the free troposphere were close to those calculated for the whole dataset, except that $\delta^{15}N^{SP}$ trends decreased after filtering out the samples with significant impact from plenary boundary layer. This indicates that N$_2$O interannual trends observed at Jungfraujoch are of regional relevance, despite the fact that a small impact from local sources can be seen. Because of the observed irregular variability and the change in sampling frequency (though no change in daily sampling time) in our dataset, we separated the time-series into two phases: April 2014-February 2016 (first phase; weekly sampling) and August 2016-December 2018 (second phase; bi-weekly sampling). In the first phase, the rates of increase in $\delta^{15}N^{SP}$ and $\delta^{18}O$ were almost one order of magnitude larger than over the whole dataset. This is most likely due to the unexpectedly low $\delta^{15}N^{SP}$ and $\delta^{18}O$ in summer 2014 followed by a distinct increase in winter 2014-2015, which results in large rates of increase over short periods. Such growth rates were not seen in the second phase, when both $\delta^{15}N^{SP}$ and $\delta^{18}O$ showed small and insignificant variations. $\delta^{15}N^{bulk}$ displayed a decreasing interannual trend in both phases; however, the rate of decrease was larger in the second phase (-0.130±0.045‰ a$^{-1}$).

We tuned our two-box model to best match the observed $N_2O$ mixing ratios and isotopic
composition at Jungfraujoch. An estimate of anthropogenic emissions and source signatures is
given in Table 4. For 2018, annual $N_2O$ emissions were estimated to be 8.6±0.6 Tg $N_2O$-N $a^{-1}$
equivalents. The average isotopic signatures for anthropogenic sources were -8.6±4‰, 34.8±3‰
and 10.7±4‰ for $\delta^{15}N^{bulk}$, $\delta^{15}N^{SP}$ and $\delta^{18}O$, respectively, which are clearly lower than those for
preindustrial $N_2O$ in the tropospheric background (Table 2; Toyoda et al., 2013).

**3.3 Seasonal variation of $N_2O$ isotopic composition**

$\delta^{15}N^{SP}$ of $N_2O$ showed the most pronounced variability among all isotopic time-series (Fig. 2),
spanning 2.5‰ for individual flask sample measurements. Seasonal variability was estimated with
the NLS model and presented as mean seasonal cycles (Fig. 3). For $\delta^{15}N^{SP}$ a "summer minimum"
was found regardless of whether the entire dataset or only the second phase was considered (Fig.
3), although seasonal variability of the second time-series was smaller and showed the minimum
occurring earlier. The seasonal pattern of $\delta^{15}N^{bulk}$ determined from the whole dataset indicates a
significant summer maximum, but this was not seen when only the data from the second phase
was taken, as there was no significant seasonal pattern over this period alone. For $\delta^{18}O$, we
observed only small temporal variability and a lack of seasonal pattern. In addition, seasonal
variations of time-series filtered for free troposphere were evaluated; these show temporal patterns
similar to the whole dataset (Fig. 3).

**3.4 Air mass origin and *in situ* measurements at Jungfraujoch**

Back-trajectory simulations indicate six major transport clusters during 2014-2018, as shown in
Figure 4a. Four of these transport regimes (SW, E, L and NW) dominate, accounting for about 60-
90% coverage of the whole period. By contrast, the free troposphere cluster only represents 10-
20% of the data. Averaged monthly contributions of transport clusters are shown in Figure 4b,
with more pronounced impact by the L, E and NW regions in summer and stronger contribution
by FT and SW in winter. The source patterns of the air masses at Jungfraujoch were generally
consistent across the years in the present study. However, an apparent discrepancy was found for
discrete sampling times in the last two years (e.g. particularly low contribution from SW) which
is most likely due to the low and variable sampling frequency of the discrete sample collection
(Fig. 4b).
The detrended seasonal variability of *in situ* measurements indicates summer maxima for $NO_y$
mixing ratios as well as $NO_y$-to-CO ratios at Jungfraujoch (Fig. 5). This likely indicates stronger
exchange with the polluted planetary boundary layer in summer (Herrmann et al., 2015; Zellweger
et al., 2003), which is consistent with the seasonal pattern of air mass footprint derived from back-
trajectory simulations. The late spring-to-summer maxima for $O_3$ mixing ratios may be attributed
to air mixing with stratosphere and/or planetary boundary layer, similar to the findings from a
previous study at Jungfraujoch (Tarasova et al., 2009). On the other hand, CO shows a maximum
in early spring and decreases in summer when its atmospheric lifetime is shortest. Atmospheric
$O_3$, $NO_y$ and CO measurements during our discrete sampling periods also well represented
seasonal variability shown for *in situ* measurements, except for 2016-2017 where there was a five-
month sampling gap (Fig. 5).
Comparisons of air mass footprints as well as $O_3$, $NO_y$ and CO mixing ratios between *in situ* and
discrete sampling indicate that the discrete sampling covers the main air source regions and
variabilities in local pollution/free troposphere fairly well (Figs. 4 and 5). In the second phase
(2016-2018), the less frequent sampling impedes evaluation of the seasonal and interannual
variabilities.

 **3.5 Relationship between $N_2O$ isotopic signatures and air mass footprints**

We categorized $N_2O$ mixing ratio and isotopic signature time-series into subsets based on the six
air mass transport clusters. One-way ANOVA among clusters indicates that $N_2O$ mixing ratios in
air masses originating from cluster L were significantly higher and those from clusters FT and W
were significantly lower than the others (Fig. 6). In accordance with the pattern found for mixing
ratios, $\delta^{15}N^{SP}$ and $\delta^{18}O$ were high for cluster FT, and low for cluster L. For $\delta^{15}N^{bulk}$, little difference
between transport clusters was detected.

## 4 Discussion

### 4.1 Quality assurance of isotopic measurements

This study reports the first results of background $N_2O$ isotopic measurements based on a laser spectroscopic technique. Benefiting from the preconcentration process, we achieved measurement repeatability for a target gas of 0.10-0.20‰ for $\delta^{15}N^{bulk}$ and $\delta^{18}O$ (Fig. S2), which is comparable to that of IRMS measurements of ambient atmosphere (Park et al., 2012; Prokopiou et al., 2017; Röckmann et al., 2003; Toyoda et al., 2013). The long-term robustness of our technique is adequate for disentangling both seasonal and interannual temporal variability as shown in Figure 2. In particular, our repeatability of target measurements for $\delta^{15}N^{SP}$ (0.15-0.25‰) appears to be better than previous studies measuring background atmosphere or firn air (0.8‰, Park et al., 2012; 0.3‰, Prokopiou et al., 2017; 0.3‰, Toyoda et al., 2013).

### 4.2 Seasonal variabilities of atmospheric $N_2O$ isotopic composition

*In situ* measurements of $N_2O$ mixing ratios showed a clear early summer maximum and late summer minimum (Fig. 1). Such a seasonal pattern was previously found for a number of NOAA and AGAGE sites analyzing long-term $N_2O$ records in the NH (Jiang et al., 2007; Nevison et al., 2011). One explanation of the late-summer minimum is a strong influence of the STE process in this period, which transports $N_2O$-depleted but isotopically enriched air downward from the stratosphere into the troposphere (Park et al., 2012; Snider et al., 2015b). During the late summer at Jungfraujoch, we find strong enrichment of $^{15}N$ in atmospheric $N_2O$ according to the detrended seasonality for the whole dataset (Fig. 3). This is supported by a FLEXPART model simulation of the contribution of upper tropospheric air to Jungfraujoch station, showing highest contributions in August (Fig. S4; Henne et al., Personal Communication). At Hateruma Island, Japan, Toyoda

et al. (2013) observed a seasonal pattern of atmospheric $N_2O$ mixing ratios which is comparable
with our study, but found insignificant variations of isotopic composition. On the other hand, $N_2O$
seasonal variability could be influenced by oceanic emission sources (Jiang et al., 2007; Nevison
et al., 2005), complicating the explanations for the observed temporal patterns. For example, in
another study looking at archived air from Cape Grim, Australia, Park et al. (2012) detected an
April-May minimum and a November-December maximum for $N_2O$. This is expected for the SH,
as STE is most prevalent in April (Nevison et al., 2011). They observed negative correlations of
$\delta^{15}N^{bulk}$, $\delta^{15}N^{\alpha}$ and $\delta^{18}O$ with $N_2O$ mixing ratios, appearing to support the idea that the STE process
is responsible for seasonal variabilities in $N_2O$ mixing ratios and isotopic composition at Cape
Grim. However, the seasonal cycle for $\delta^{15}N^{\alpha}$ was much larger than $\delta^{15}N^{bulk}$ and $\delta^{18}O$, which could
not be explained by STE alone. They suggested that the seasonal patterns of $N_2O$ isotopes at Cape
Grim may be due to mixing between oceanic sources (high $N_2O$ with low $^{15}N$ and $^{18}O$) and STE
(low $N_2O$ with high $^{15}N$ and $^{18}O$) (Nevison et al., 2011; Park et al., 2012). However, because we
observed a concurrent minimum of $\delta^{15}N^{SP}$ and maximum of $\delta^{15}N^{bulk}$ in July-August with low $N_2O$
at Jungfraujoch (Fig. 3), additional mechanisms must be considered here.
Regional model simulations based on Swiss $N_2O$ emissions derived from the inverse method were
used to explore contributions from different sources to the variability in $N_2O$ enhancements at
Jungfraujoch. As shown in Figure 7a&7b, soil emissions, including direct and indirect emissions
from agricultural lands and emissions from (semi-)natural areas, account for more than 70% of the
total $N_2O$ enhancements, while manure and waste management contribute another 20%. Total $N_2O$
enhancements appeared to be highest in May to July (Fig. 7c), in accordance with the highest
contribution by soil emissions. The early-to-middle summer maximum in the simulated $N_2O$
enhancements is comparable with maximum of $N_2O$ mixing ratios in early summer as observed at
Jungfraujoch (Fig. 1c). This underlines the importance of soil emission in accounting for
atmospheric $N_2O$ variability (Saikawa et al., 2014). In late summer, the minimum of $\delta^{15}N^{SP}$ (Fig.
3) may be then attributed to the influence of soil emitted $N_2O$, which has lower $\delta^{15}N^{SP}$ (7.2±3.8‰;
Table 1) than the troposphere (Fig. 2). However, the STE process, which resulted in the minimum
of $N_2O$ mixing ratio, likely contributes a positive isotope effect in the meanwhile (Kaiser et al.,
2006). In order to evaluate the combined effect of STE and soil emission on $\delta^{15}N^{SP}$ in late summer,
we applied a mixing calculation. Such estimate was made based on the approximated $N_2O$
enhancement/depletion contributed by the two processes and the assumed isotope effects (see more
details in M&M). The mixing calculation indicated an overall isotope effect of about 0.01‰, which
is extremely small and below our analytic precision. This practice suggests that it is still
challenging to build a direct link between $N_2O$ sources/processes and the observed isotope
signature in the background atmosphere. It is also noteworthy that the $\delta^{15}N^{SP}$ used in the
calculation (7.2±3.8‰) may underestimate the isotope effects of soil emission, given that
denitrification, as a major $N_2O$ process in soils, produces $N_2O$ with $\delta^{15}N^{SP}$ close to 0‰ (Sutka et
al., 2006). Previous field studies have demonstrated that low-$\delta^{15}N^{SP}$ $N_2O$ emissions (~0‰), i.e.
following the denitrification pathway, predominates during summer periods at Swiss (Wolf et al.,
2015) and German (Ibraim et al., 2019) grasslands. By contrast, the influence of biogeochemical
processes (nitrification and denitrification) on $\delta^{15}N^{bulk}$ is generally smaller than that on $\delta^{15}N^{SP}$
(Toyoda et al., 2011), and such effect on $\delta^{15}N^{bulk}$ are usually overwritten by the wide range of
isotopic signatures in soil N substrates (Sutka et al., 2006). Hence, given the distinct $\delta^{15}N^{bulk}$
maximum and $N_2O$ minimum in late summer during our observation (Figs. 1 and 3), we suggest
that the STE process is mainly responsible for the seasonal variability in $\delta^{15}N^{bulk}$.
The footprint analyses based on air mass residence time revealed a seasonal pattern, with a higher
contribution of background air from the FT and SW regions in winter and more pronounced
contribution of local planetary boundary layer air from the L, E and NW regions in summer (Fig.
4b). The higher frequency of air mass footprints recently in contact with the surface in summer is
consistent with inverse modeling results, indicating a larger contribution of soil $N_2O$ emissions in
June/July (Fig. 7). For the air mass regime representing the free troposphere, $N_2O$ mixing ratios
observed at Jungfraujoch were significantly below the average, while $\delta^{15}N^{SP}$ and $\delta^{18}O$ were higher
(Fig. 6). By contrast, the local cluster (L) representing a strong impact from the planetary boundary
layer had higher $N_2O$ mixing ratios and lower isotopic signatures (except $\delta^{15}N^{bulk}$) than the other
source regions. In addition, the ratio of $NO_y$ to CO, which is a more straightforward indicator of
the free troposphere (Zellweger et al., 2003), show significant negative correlations with $\delta^{15}N^{SP}$
and $\delta^{18}O$, but not with $\delta^{15}N^{bulk}$ (Fig. 8). This further suggests that the seasonal variability of $\delta^{15}N^{SP}$
and $\delta^{18}O$ observed at Jungfraujoch is most likely influenced by ground-derived emissions, while
fluctuations in $N_2O$ mixing ratios and $\delta^{15}N^{bulk}$ are possibly driven by STE.
Considering the complexity in mechanisms responsible for $N_2O$ isotopic variations, we strongly
recommend more field measurements of $N_2O$ isotopic signatures at higher frequency and at
different background sites, in order to cover spatial and temporal variability in $N_2O$ sources. For
example, in the second phase, we only detected a significant seasonality of $\delta^{15}N^{SP}$, with a minimum
in July, which is one month earlier than the summer minimum found for the whole dataset (Fig.
3). This may be attributed to a difference in source regions, as Northwest regions appeared to be
significantly more important during 2017 (second phase). However, due to low sampling
frequency, it is challenging to overcome the large uncertainty in seasonality analysis for a two-
year period such as the second phase.
Based on our bottom-up approach, we simulated isotopic signatures for the overall $N_2O$ sources
responsible for the N₂O mixing ratio increase in the atmosphere (Fig. 9). However, the
interpretation of simulated versus observed variability in N₂O isotopic composition was difficult,
except for the somewhat similar patterns in $\delta^{18}O$. Our results suggest a limitation in the current
knowledge and literature values on isotopic signatures of most N₂O sources. In addition, most N₂O
sources may not exhibit a well-defined isotopic signature but a range of values regulated under a
number of processes/environmental factors. For example, isotopic signatures of soil-derived N₂O
are often determined by an interaction of several soil and climatic factors. It might be possible in
the future to model these changes implementing isotopes in ecosystem models, as recently
demonstrated by Denk et al. (2019).
**4.3 Interannual trends of atmospheric N₂O isotopic composition**
Over a period of almost five years, our observations show an interannual increase in N₂O mixing
ratio and decrease in $\delta^{15}N^{bulk}$ (Fig. 10). This is to be expected, assuming that the atmospheric N₂O
increase is primarily attributed to anthropogenic sources, which emit isotopically lighter N₂O
relative to the tropospheric background (Table 1) (Rahn and Wahlen, 2000). Compared to several
studies on firn air (Ishijima et al., 2007; Röckmann et al., 2003) and surface air (Park et al., 2012;
Röckmann and Levin, 2005; Toyoda et al., 2013), the rate of decrease for $\delta^{15}N^{bulk}$ at Jungfraujoch
is relatively high (-0.05 to -0.06 ‰ a⁻¹, Table 3). Such a discrepancy in the $\delta^{15}N^{bulk}$ trend could be
due to a large contribution of terrestrial N₂O emission from the European continent to Jungfraujoch
(Figs. 6 and 7), as N₂O originating from soil emissions is significantly more isotopically depleted
than that of oceanic sources (Snider et al., 2015b). Nevertheless, our observation period is shorter
than that of other studies, so the interannual trends determined here are more likely affected by
year-to-year variability. Among all reported records, the decrease of $\delta^{15}N^{bulk}$ observed at Hateruma
Island was the most up-to-date and smallest (-0.020-0.026‰ a⁻¹) (Toyoda et al., 2013). The authors
argued that the smaller declining trend for $\delta^{15}N^{bulk}$ may be explained by the recent increase in
anthropogenic isotopic ratios particularly for agricultural $N_2O$ emissions, although Ishijima et al.
(2007) suggested a decline in both $\delta^{15}N^{bulk}$ and $\delta^{18}O$ in anthropogenic $N_2O$ from 1952-1970 to
1970-2001 based on inverse modeling.
For the interannual trends observed at Jungfraujoch, it is noteworthy to point out that our
observations covering a rather short period may lead to large uncertainties despite statistical
significance. The discrepancy found in the trends between the first and second phases indicates
that variability of $N_2O$ isotopic composition is likely to obscure interannual trends over shorter
periods (Toyoda et al., 2013). Hence, extended time-series of isotopic measurements are needed
to reevaluate, for example, the observed tendency of increase in $\delta^{18}O$ and $\delta^{15}N^{SP}$ at Jungfraujoch
(Table 3; only significant during the first phase). For $\delta^{18}O$ of atmospheric $N_2O$, a generally
declining trend smaller than that of $\delta^{15}N^{bulk}$ has been indicated by a number of observations
(Bernard et al., 2006; Ishijima et al., 2007; Park et al., 2012; Röckmann et al., 2003; Röckmann
and Levin, 2005). This is expected as $\delta^{18}O$ of anthropogenic $N_2O$ is not much different from that
of the natural background, assuming that the oxygen atom in $N_2O$ is largely derived from soil water
and ambient oxygen during production (Rahn and Wahlen, 2000).
It is still a challenging task to disentangle interannual trends of $\delta^{15}N^{SP}$-$N_2O$ in the background
atmosphere, due to limitations in analytical repeatability and precision (Harris et al., 2017; Mohn
et al., 2014). Past results have reached inconsistent conclusions, showing positive (Bernard et al.,
2006; Park et al., 2012; Prokopiou et al., 2017; Röckmann and Levin, 2005) or negative
(Röckmann et al., 2003; Toyoda et al., 2013) trends of similar magnitude (Fig. 10). On the one
hand, the negative trend in $\delta^{15}N^{SP}$ could be explained by the significantly lower $\delta^{15}N^{SP}$ from
anthropogenic sources (e.g. agricultural sources; Table 1) than of the tropospheric background
(near 18‰; Fig. 10). On the other hand, Park et al. (2012) suggested that the increase of $\delta^{15}N^{SP}$ in
the atmospheric $N_2O$ may reflect a global increase in importance of the contribution by nitrification
(high- $\delta^{15}N^{SP}$ process) to agricultural $N_2O$ emissions. This is based on the assumption that the
growth of $N_2O$ emissions is largely due to enhanced fertilizer application which promotes
nitrification activity (Pérez et al., 2001; Tian et al., 2019). The observed mean increase rate of 0.02‰
$a^{-1}$ for $\delta^{15}N^{SP}$ by Park et al. (2012) could then be translated into an increase of 13-23% for the
relative amount of nitrification-derived $N_2O$ between 1750 and 2005. However, this should be
further evaluated with more frequent sampling (Park et al. (2012) only sampled 1-6 times per year)
and tested with isotopic measurements across the NH, where agricultural $N_2O$ emissions are more
dominant than in the SH. In addition, the strong seasonal pattern of $\delta^{15}N^{SP}$ at Jungfraujoch suggests
that seasonal variations of $\delta^{15}N^{SP}$ in response to climatic or source factors are crucial and must be
taken into consideration for evaluating interannual $\delta^{15}N^{SP}$ trends.
**4.4 Simulated anthropogenic $N_2O$ sources with the two-box model and comparison with**
**other studies**
To further evaluate anthropogenic source signatures of $N_2O$ isotopic composition, we applied a
two-box model representing a well-mixed troposphere and stratosphere (Röckmann et al., 2003;
Schilt et al., 2014; Sowers et al., 2002). The model runs with the whole dataset and the dataset
filtered for free-troposphere only (Table 4) exhibit statistically identical results, supporting that
our model estimates, with observations at Jungfraujoch, are representative of the background
atmosphere. The simulated trends of the $N_2O$ mixing ratios and isotopic composition show a
gradual increase in $N_2O$ and decrease in the isotopic signatures (see Fig. 10), which agree with
existing observations within the model uncertainty. However, this does not hold for individual
studies considered separately. For example, the $N_2O$ mixing ratios observed by Röckmann et al.
(2003) and Prokopiou et al. (2017) would lead to a higher preindustrial $N_2O$ compared to our
model simulation, which is likely due to the uncertainty in the firn air records (Prokopiou et al.,

632 2017).

We compared the anthropogenic isotopic signatures determined by our two-box model with other
similar studies in Table 4. Our estimates generally lie within the ranges given in the earlier studies
(Ishijima et al., 2007; Park et al., 2012; Prokopiou et al., 2017; Röckmann et al., 2003; Sowers et
al., 2002; Toyoda et al., 2013). However, isotopic signatures of $N_2O$ sources estimated for 2018 in
this study are higher in $\delta^{15}N^{bulk}$ and $\delta^{18}O$ (by 4-8‰), and lower in $\delta^{15}N^{SP}$ (by 2-7‰) than model
estimates for the early 2000s from two other studies from SH (Park et al., 2012; Prokopiou et al.,
2017). Such differences in $\delta^{15}N^{bulk}$ and $\delta^{18}O$ could be related to interhemispheric differences, as
the relative contributions of $N_2O$ sources vary between the two hemispheres (Toyoda et al., 2013).
Also, more interestingly, this could suggest a shift in the $N_2O$ source isotopic signatures over the
last few decades. For example, an increase of $\delta^{15}N^{bulk}$ in anthropogenic $N_2O$ sources over time
may be attributed to growing contributions of other industrial/waste sources with high $\delta^{15}N^{bulk}$
(Prokopiou et al., 2017). In addition, if the assumption of increasing $\delta^{15}N^{bulk}$ and decreasing $\delta^{15}N^{SP}$
in anthropogenic $N_2O$ sources over time holds, it points to a recently growing contribution of
denitrification relative to nitrification, to the global atmospheric $N_2O$ increase (Sutka et al., 2006;
Toyoda et al., 2013). By contrast, Park et al. (2012) and Prokopiou et al. (2017) proposed an
increasing importance of nitrification for anthropogenic $N_2O$ emissions based on the increasing
$\delta^{15}N^{SP}$ trend since 1940. This may suggest that a strong climate change feedback has recently
resulted in significant shifts in $N_2O$ source process, hence twisting the isotopic signatures of
anthropogenic sources (Griffis et al., 2017; Xu-Ri et al., 2012). Alternatively, the uncertainty in
determining $N_2O$ isotopic signatures in the background atmosphere and inter-laboratory
comparability may play a role in the observed discrepancy.
Given the strong heterogeneity in source contributions to $N_2O$ emissions around the globe
(Saikawa et al., 2014), current two- and four-box model estimates based on observations at
individual sites or regions are likely to reflect latitudinal or even interhemispheric differences in
anthropogenic isotopic signatures. On the other hand, previous discussions of the model
sensitivities by Röckmann et al. (2003) and Toyoda et al. (2013) have suggested that anthropogenic
isotopic values are most sensitive to the trends in tropospheric isotopic values and the relative
difference in tropospheric isotopic values between present and preindustrial times. For example,
given the similar parameters used for preindustrial times as our study, Park et al. (2012) observed
much lower $\delta^{15}N^{bulk}$ in the recent troposphere than in our case, hence resulting in significantly
lower $\delta^{15}N^{bulk}$ for the anthropogenic source. Furthermore, Park et al. (2012) and Prokopiou et al.
(2017) simulated a positive trend in $\delta^{15}N^{SP}$ relative to preindustrial times, which in return resulted
in a much higher $\delta^{15}N^{SP}$ for the anthropogenic sources.
Using an alternative bottom-up approach, we estimated the anthropogenic source isotopic
signatures based on the $N_2O$ emission inventory simulated for Jungfraujoch and published source
isotopic signatures as summarized by Harris et al. (2017) (Table 1). The retrieved anthropogenic
isotopic signatures (Table 5) were largely in agreement with the isotopic signature of agricultural
soil emissions (Snider et al., 2015b; Wolf et al., 2015), indicating that this source could explain
more than 60% of the total $N_2O$ emissions. However, the anthropogenic isotopic signatures
estimated by this approach were lower than the results from our two-box model (Table 4). In
contrast, another similar bottom-up estimate based on the global $N_2O$ emission inventory (Toyoda
et al., 2013) reported anthropogenic isotopic values that agree well with our box-model results.
This may be explained by the different isotopic signatures used to describe agricultural $N_2O$
emissions, as those values used for the bottom-up estimates by Toyoda et al. (2013) were
significantly lower (Toyoda et al., 2011) than those used in this study (Snider et al., 2015b; Wolf
et al., 2015). Such bottom-up estimation suggests that more isotopic measurements of the
background atmosphere from different regions, and better constraints on individual anthropogenic
(especially agricultural) $N_2O$ isotopic signatures, are necessary for a better representation of $N_2O$
isotopic composition in atmospheric modeling studies.

## 5 Conclusions

With the recently developed laser spectroscopic technique coupled with a preconcentration device, we achieved good repeatability in measurements of $N_2O$ isotopic composition from the background atmosphere at Jungfraujoch, Switzerland. This time-series covered a period of five years and showed a distinct seasonality, with $\delta^{15}N^{bulk}$ maxima and $\delta^{15}N^{SP}$ minima in late summer, associated with the lowest $N_2O$ mixing ratios over the year. The seasonal fluctuation of $\delta^{15}N^{bulk}$ was associated with the stratosphere-troposphere exchange process, in agreement with other monitoring networks (Nevison et al., 2011), while the contrasting depletion of $\delta^{15}N^{SP}$ in later summer is possibly a combined result of STE and agricultural emissions, with the latter being more important. The analyses of air mass transport regimes together with the simulation of $N_2O$ enhancements for Jungfraujoch supported our explanations and highlighted that the fluctuation between the free troposphere and local contributions dominated by soil emission drives the seasonality of $\delta^{15}N^{SP}$ and $\delta^{18}O$ as observed at Jungfraujoch.

We found statistically significant interannual trends for $\delta^{15}N^{bulk}$, which is expected as anthropogenic $N_2O$ sources are characterized by low $^{15}N$ abundance. For $\delta^{15}N^{SP}$ and $\delta^{18}O$, interannual trends were highly uncertain and possibly masked by their large temporal variabilities. Using a two-box model approach, we simulated the evolution of $N_2O$ isotopic composition from preindustrial times to the present. This model suggests an overall decreasing trend for all isotopic deltas in conjunction with the atmospheric $N_2O$ increase. The anthropogenic source signatures given by the model generally agreed with previous studies. However, these model results are still sensitive to the ranges and trends of the observed $N_2O$ isotopic signatures in the present troposphere. In the future, more extended records of high-precision $N_2O$ isotopic measurements

704 and application of multiple-box modeling approaches (Rigby et al., 2013) are necessary to account

705 for the global $N_2O$ budget and evolution of anthropogenic sources.

**Data availability**

Data for this study have been deposited in a general data repository (https://figshare.com/s/077562ab408dd1bd0880; doi:10.6084/m9.figshare.12032760.v1, 2020).

**Author contribution:**

LY, EH and JM led and designed this study. LY, EH, SE conducted sample collection at Jungfraujoch; LY and EH analyzed discrete samples at Empa; MS and CZ contributed *in situ* measurements of $N_2O$, $NO_y$, CO and $O_3$ at Jungfraujoch; LY, EH and SH performed data analyses for the time-series and conducted model simulations. LY wrote the main manuscript; EH, SH and JM were involved in the revisions of the manuscript and commenting. SE, MS, LE and CZ were also involved in scientific discussion and commenting on the manuscript.

**Competing interests**

The authors declare that they have no conflict of interest.

**Acknowledgements**

We are thankful to the research infrastructure provided by the High Altitude Research Stations Jungfraujoch and Gornergrat. We are grateful to the help from the custodians (Mr. and Mrs. Fischer and Mr. and Mrs. Käser) at the research station of Jungfraujoch. We would like to thank Simon Wyss, Kerstin Zeyer, Patrik Zanchetta and Flurin Dietz for their support with the sample collection as well as laboratory assistance. The NABEL network is operated by Empa in collaboration with the Swiss Federal Office for the Environment. Prof. Sakae Toyoda and Prof. Naohiro Yoshida from Tokyo Institute of Technology are acknowledged for their analyses of the applied reference standards. This study was financially supported by the Swiss National Science Foundation (grant number 200021_163075) and the Swiss contribution to the Integrated Carbon Observation System

(ICOS) Research Infrastructure (ICOS-CH). ICOS-CH is funded by the Swiss National Science
Foundation and in-house contributions. Longfei Yu was additionally supported by the
EMPAPOSTDOCS-II program, which receives funding from the European Union's Horizon 2020
research and innovation program under the Marie Skłodowska-Curie grant agreement number

732 754364.

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

1032    **Table 1** An overview of N$_2$O emission sectors for Swiss Meteotest Inventory and global EDAGR Inventory and available source isotopic
1033    signatures (‰)*

| Meteotest Category | Meteotest Source(s) | EDGAR Category[φ] | EDGAR Primary source(s)[φ] | $\delta^{18}O$ | $\delta^{15}N^{bulk}$ | $\delta^{15}N^{SP}$ | References |
|---|---|---|---|---|---|---|---|
| Orgs | Organic soils | 7B, 7C | Indirect soil emissions | 29.0±3.7 | -17.8±5.7 | 7.2±3.8 | Snider et al. (2015b), Wolf et al. (2015) |
| IndustrialHeating | Cement production, industrial combustion, furnaces, waste incinerator, other industrial | 1A1, 1A2 | deNO$_x$ use in fossil fuel and MSW incineration plants | 35.9±13.1 | 3.9±2.9 | 17.6±0.5 | Ogawa et al. (2005a), Harris et al. (2015) |
| Transport | Agricultural and construction machinery, road traffic | 1A3a, 1A3c, 1A3d, 1A3e, 1A3b | Fuel combustion in non-road transportation | 28.6±9.9 | -28.7±3.6 | 4.2±2.4 | Toyoda et al., (2008) |
| | | | Fuel combustion for road transportation | 40.3±3.7 | -7.2±1.2 | 10.0±4.3 | Toyoda et al., (2008) |
| Heating | Agricultural, commercial and private heating | 1A4 | Fuel combustion: other sectors (dominantly household heating) | 37±10 | 5.5±6 | 3.5±4 | Ogawa et al. (2005a), Ogawa et al. (2005b) |
| Refinery | Refineries | 1B2a | Refineries | - | - | - | - |
| IndustryAndUse | Nitric acid production, use in households and hospitals | 2 and 3 | Nitric acid production (adipic acid, medical, and private (aerosol) use) | 29.1±18.8 | -8.3±10.6 | 3.3±5.5 | Toyoda et al., (2008), Thiemens et al. (1991) |
| Manure | Manure management | 4B | Manure management | 23.9±3.8 | -17.5±6.2 | 6.5±4.1 | Maeda et al. (2010) |
| DirectAgri | Crop residues/soil organic matter, animal waste on pastures, synthetic fertilizer use, manure application | 4C, 4D | Direct soil emissions | 29.0±3.7 | -17.8±5.7 | 7.2±3.8 | Snider et al. (2015b), Wolf et al. (2015) |
| IndirectAgri | Leaching, other indirect emissions from agri. Soils | 4D3 | Direct soil emissions | 29.0±3.7 | -17.8±5.7 | 7.2±3.8 | Snider et al. (2015b), Wolf et al. (2015) |

| WastBurning | Illegal waste burning | 4F | Agricultural waste burning | 25±3.0 | -1.0±3.0 | 2.8±3.0 | Ogawa et al. (2005b) |
|---|---|---|---|---|---|---|---|
| Waste | Industrial fermentation, wastewater treatment, sewage sludge burning | 6 | Waste (wastewater treatment) | 31.5±14.1 | -11.6±12.7 | 10.5±5.7 | Snider et al. (2015b) |
| IndirectNat | Indirect emissions from (semi-)natural ecosystems | 7B, and 7C | Indirect soil emissions | 29.0±3.7 | -17.8±5.7 | 7.2±3.8 | Snider et al. (2015b), Wolf et al. (2015) |

1034 *Isotopic signatures for anthropogenic sources are obtained from the summary by Harris et al. (2017).

1035 φ These are the primary sources contributing to $N_2O$ emissions in Switzerland

 **Table 2** Input variables for simple two-box model

| Variable | Description | Value | Error distribution | References |
|---|---|---|---|---|
| $m_{trop}$ | Air in the troposphere (mol) | $1.5 \times 10^{20}$ | Constant | Röckmann et al. (2003), Schilt et al. (2014) |
| $m_{strat}$ | Air in the stratosphere (mol) | $2.7 \times 10^{19}$ | Constant | Röckmann et al. (2003), Schilt et al. (2014) |
| $F_{ex}$ | Troposphere-stratosphere exchange rate (kg a$^{-1}$) | $(5.37\pm1.26) \times 10^{17}$ | Uniform | Röckmann et al. (2003), Schilt et al. (2014) |
| $F_{ocean}$ | Oceanic N$_2$O flux (Tg a$^{-1}$ N equivalents) | $4\pm1$ | Gaussian | Schilt et al. (2014) |
| $\tau_{PI}$ | Preindustrial N$_2$O lifetime (year) | $123\pm10$ | Gaussian | Prather et al. (2015) |
| $\tau_{PD}$ | Present day N$_2$O lifetime (year) | $116\pm9$ | Gaussian | Prather et al. (2015) |
| $c_{PI}$ | Mixing ratio in the preindustrial troposphere (nmol mol$^{-1}$) | $270\pm7.5$ | Uniform | Sowers et al. (2002), Röckmann et al. (2003) |
| $\delta^{15}N^{bulk}{}_{PI}$ | Mean $\delta^{15}N^{bulk}$ of preindustrial tropospheric N$_2$O (‰) | $8.9\pm2$ | Gaussian | Toyoda et al. (2013) |
| $\delta^{18}O_{PI}$ | Mean $\delta^{18}O$ of preindustrial tropospheric N$_2$O (‰) | $46.1\pm2$ | Gaussian | Toyoda et al. (2013) |
| $\delta^{15}N^{SP}{}_{PI}$ | Mean $\delta^{15}N^{SP}$ of preindustrial tropospheric N$_2$O (‰) | $19.05\pm2$ | Gaussian | Toyoda et al. (2013) |
| $\delta^{15}N_{ocean}$ | Mean $\delta^{15}N^{bulk}$ for oceanic emissions (‰) | $5.1\pm1.9$ | Uniform | Snider et al. (2015b) |
| $\delta^{18}O_{ocean}$ | Mean $\delta^{18}O$ for oceanic emissions (‰) | $44.8\pm3.6$ | Uniform | Snider et al. (2015b) |
| $\delta^{15}N^{SP}{}_{ocean}$ | Mean $\delta^{15}N^{SP}$ for oceanic emissions (‰) | $15.8\pm7.1$ | Uniform | Snider et al. (2015b) |
| $\delta^{15}N^{bulk}{}_{terr}$ | Mean $\delta^{15}N^{bulk}$ for emissions from terrestrial soils (‰) | $-16.7\pm11.2$ | Uniform | Snider et al. (2015b) |
| $\delta^{18}O_{terr}$ | Mean $\delta^{18}O$ for emissions from terrestrial soils (‰) | $30.1\pm9.6$ | Uniform | Snider et al. (2015b) |
| $\delta^{15}N^{SP}{}_{terr}$ | Mean $\delta^{15}N^{SP}$ for emissions from terrestrial soils (‰) | $10.1\pm11.2$ | Uniform | Snider et al. (2015b) |

1037

**Table 3** Trends of amospheric $\delta^{15}N^{bulk}$, $\delta^{15}N^{SP}$ and $\delta^{18}O$ at Jungfraujoch determined using discrete measurements between April 2014 and December 2018. The trends are determined for the whole dataset, the dataset filtered for free troposphere (removing data points with significant influence from plenary boundary layer) and the second-phase dataset with bi-weekly measurements (August 2016 to December 2018).

| | $\delta^{15}N^{bulk}$ (‰ a$^{-1}$) | | $\delta^{15}N^{SP}$ (‰ a$^{-1}$) | | $\delta^{18}O$ (‰ a$^{-1}$) | |
|---|---|---|---|---|---|---|
| | Raw | Deseasonalized | Raw | Deseasonalized | Raw | Deseasonalized |
| Whole dataset | -0.059±0.012* | -0.052±0.012* | 0.069±0.029 | 0.065±0.027 | 0.020±0.011 | 0.019±0.011 |
| Free troposphere | -0.060±0.014* | -0.054±0.013* | 0.054±0.034 | 0.036±0.030 | 0.024±0.013 | 0.019±0.011 |
| First phase (Apr. 2014-Feb. 2016) | -0.036±0.038 | -0.041±0.035 | 0.449±0.100* | 0.314±0.082* | 0.238±0.029* | 0.207±0.026* |
| Second phase (Aug. 2016-Dec. 2018) | -0.105±0.049 | -0.130±0.045* | 0.028±0.067 | -0.007±0.066 | -0.007±0.042 | -0.001±0.040 |

* Indicate significance of linear regression.

**Table 4** Results of the two-box model simulations and selected literature values for comparison.

| | This study | | $RMSE^{\varphi}$ | Sowers et al. (2002)[a] | Röckmann et al. (2003)[b] | Ishijima et al. (2007)[c] | Toyoda et al. (2013)[d] | Park et al. (2012)[e] | Prokopiou et al. (2017)[f] |
|---|---|---|---|---|---|---|---|---|---|
| Air sample age | $NH^{\dagger}$ | $NH(FT^{\eta})$ | | $FA^{\dagger}, IC^{\dagger}$ | FA | FA | NH | $SH^{\dagger}$, FA | FA |
| | 2014-2018 | 2014-2018 | | 1785-1995 | NA | 1960-2001 | 1999-2010 | 1940-2005 | 1940-2008 |
| $\alpha*$ | 0.0154±0.004 | 0.0154±0.004 | 0.65 nmol mol$^{-1}$ | 0.0111 to 0.0128 | NA | NA | NA | NA | NA |
| $F_{anth,2018}$ (Tg N a$^{-1}$) | 8.6±0.6 | 8.5±0.6 | $NA^{\dagger}$ | 4.2 to 5.7 | 6.9 | NA | 5.5 | 6.6 | 5.4±1.7 |
| $\delta^{15}N^{bulk}$-anth (‰) | -8.6±4 | -8.5±4 | 0.23 | -7 to -13 | -11.4 | -11.6 | -9.84 | -15.6±1.2 | -18.2±2.6 |
| $\delta^{18}O$-anth (‰) | 34.8±3 | 34.3±3 | 0.22 | 17 to 26 | 31.7 | NA | 35.95 | 32.0±1.3 | 27.2±2.6 |
| $\delta^{15}N^{SP}$-anth (‰) | 10.7±4 | 10.7±4 | 0.50 | NA | 11.3 | NA | 8.52 | 13.1±9.4 | 18.0±8.6 |

[†] NH and SH: surface atmosphere from the Northern and Southern Hemisphere, respectively; FA: firn air; IC: ice core air; NA: not available.
[η] FT: Jungfraujoch dataset filtered for free troposphere (based on $NO_y$:CO).
* "Value" is the dimensionless constant $\alpha$ describing the exponential increase in the anthropogenic flux
[φ] RMSE refers to root mean square error. It is in nmol mol$^{-1}$ for $\alpha$, referring to the present day tropospheric mixing ratio for $N_2O$. For source isotopic values, RMSE is in the unit of ‰. Simulations with the whole dataset and the dataset filtered for free troposphere yielded the same RMSE.
[a] Estimates are for 1995.
[b] Estimates are for 1998; isotopic signatures of anthropogenic sources were calculated assuming modern tropospheric values to be the same as this study.
[c] Estimate is for 2000, for $\delta^{18}O$ calibration is not comparable.
[d] Estimates are for 2012 using the "Base" scenario.
[e] Estimates are for 2005.
[f] $\delta_{anth}$ values are averaged values for the period of 1940-2008.

**Table 5** Isotopic signatures for the overall, anthropogenic and major $N_2O$ sources contributing to $N_2O$ variations at Jungfraujoch. Source signatures were estimated based on a "bottom-up" approach, with literature-derived isotopic signatures and fluxes for variable sources under the Swiss Meteotest emission inventory.

| | Emission inventory (%) | $\delta^{15}N^{bulk}$ (‰) | $\delta^{15}N^{SP}$ (‰) | $\delta^{18}O$ (‰) | Reference |
|---|---|---|---|---|---|
| Overall source | 100 | -15.8 (6.2) | 7.3 (3.9) | 29.4 (5.5) | - |
| Anthropogenic source | 89.4 | -15.6 (6.3) | 7.4 (4.0) | 29.5 (5.7) | - |
| Agricultural emission | 61.5 | -17.8 (5.7) | 7.2 (3.8) | 29.0 (3.7) | Snider et al. (2015) Wolf et al. (2015) |
| Manure management | 7.4 | -17.5 (6.2) | 6.5 (4.1) | 23.9 (3.8) | Maeda et al. (2010) |
| Waste[*] | 7.2 | -11.5 (12.6) | 10.4 (5.7) | 31.3 (14.0) | Ogawa and Yoshida (2005) Snider et al. (2015) |
| Natural emission | 10.9 | -17.8 (5.7) | 7.2 (3.8) | 29.0 (3.7) | Snider et al. (2015) Wolf et al. (2015) |

[*] "Waste" sources consist of both wastewater treatment and agricultural waste burning (biomass burning).

# Figures

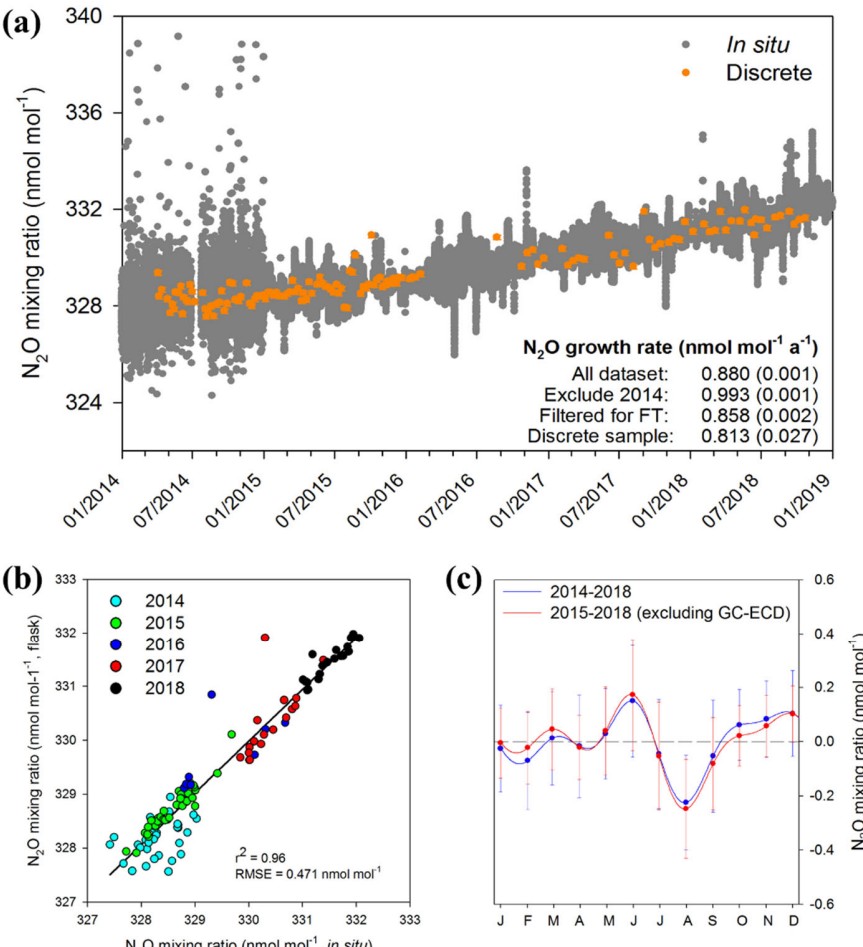

1065

**Figure 1a** *In situ* (10-min averages) and discrete measurements of $N_2O$ mixing ratios from April 2014 to December 2018 at Jungfraujoch. *In situ* $N_2O$ mixing ratio measurements were performed with GC-ECD method between April and December 2014. After that, OA-ICOS became the major analytical method for *in situ* measurements. Discrete sample points are presented as averages with error bars (one standard deviation). Annual $N_2O$ growth rates determined by linear regression are given in the figure (uncertainty shown as one standard deviation). A sampling gap exists for discrete samples between February and August 2016.

**1b** Comparison of in situ and discrete measurements of $N_2O$ mixing ratios; in situ measurements were 10-minute values averaged over the exact period of discrete sampling time (~ 40 min); in situ measurements were performed with GC-ECD method in 2014, and this was replaced with OA-ICOS method from January 2015.

**1c** Seasonality of $N_2O$ mixing ratios at Jungfraujoch derived from *in situ* measurements. Datasets with/without GC-ECD measurements are compared for seasonality evaluation. The NLS model simulation for time-series gives the detrended seasonality, with error bars indicating one standard deviation of monthly residuals.

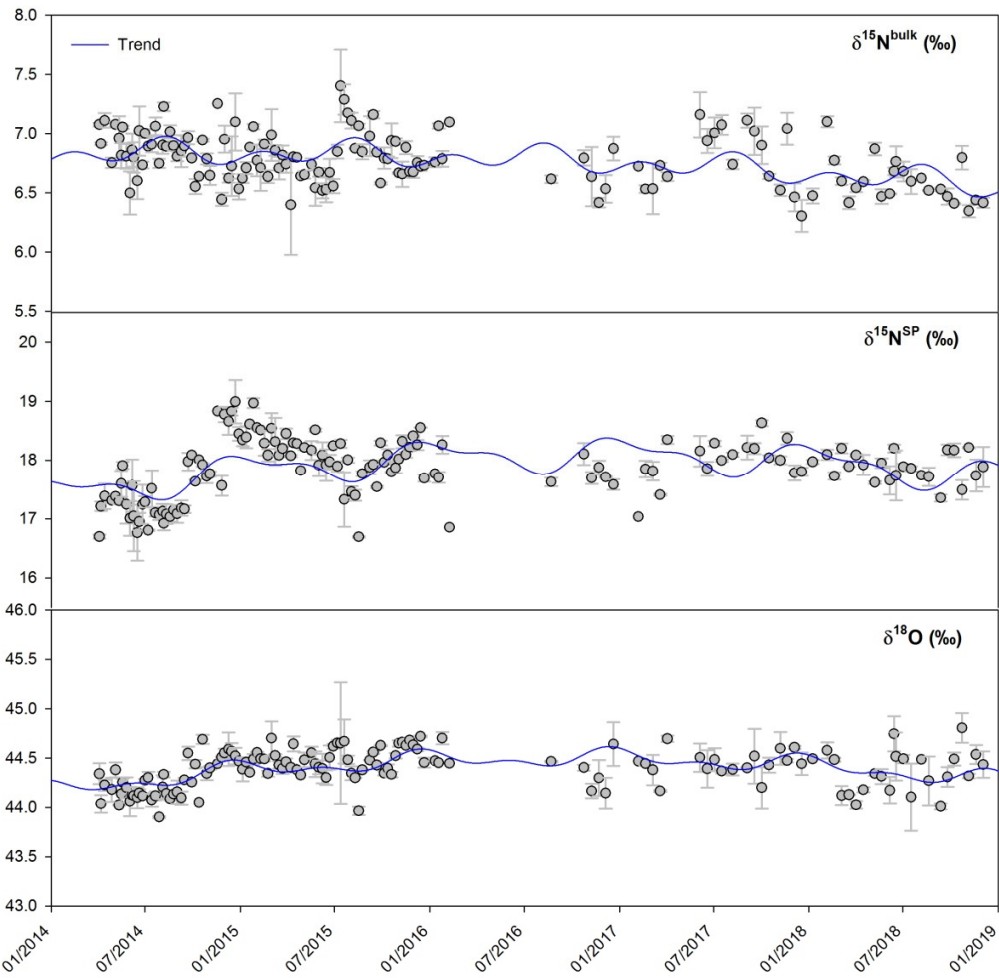

**Figure 2** Time-series of isotopic composition of atmospheric N₂O observed at Jungfraujoch from April 2014 to December 2018. Error bars indicate one standard deviation of repeated measurements. Blue lines indicate the simulated trends by the NLS model.

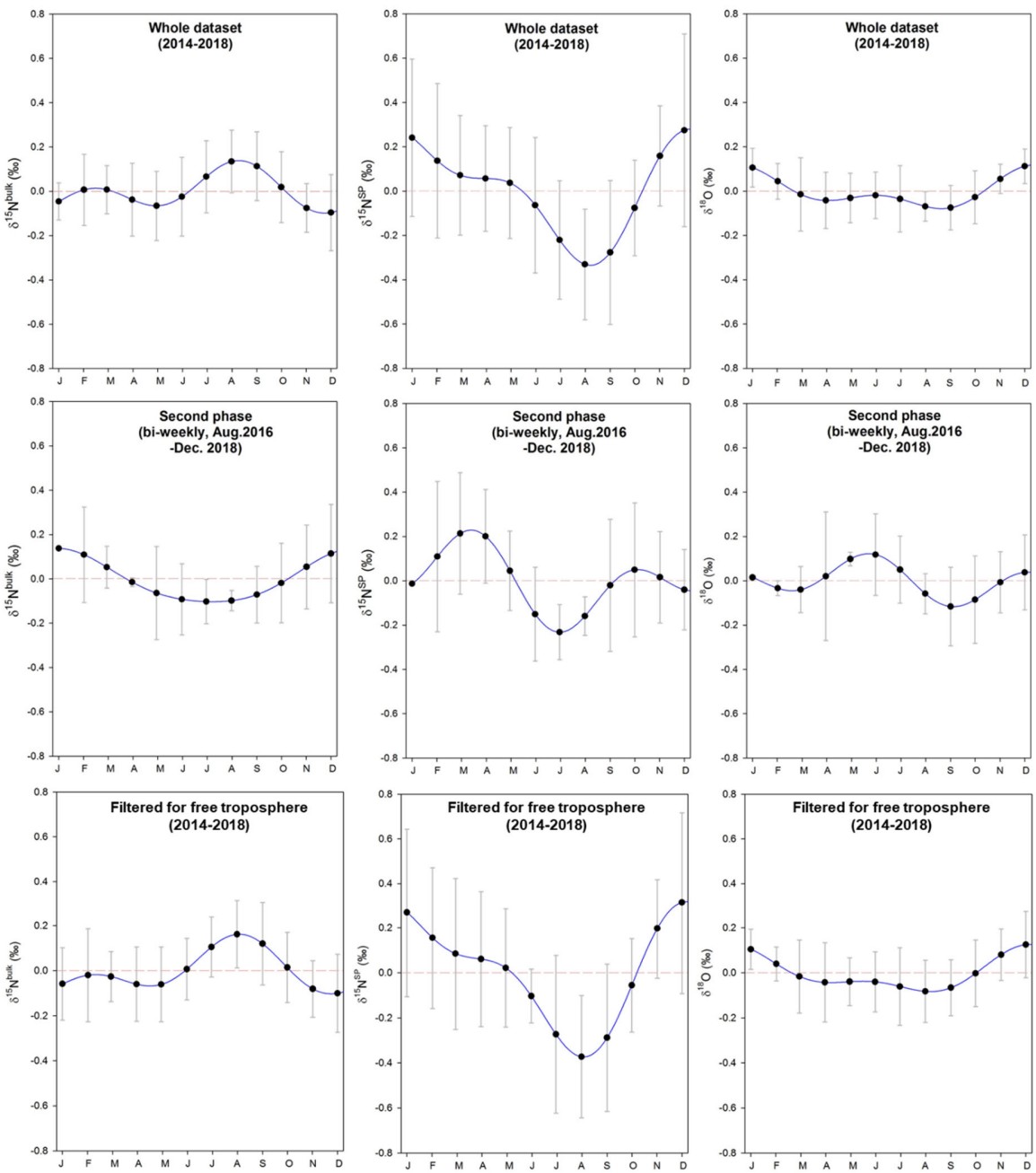

**Figure 3** Seasonality of isotopic signatures of atmospheric N$_2$O observed at Jungfraujoch. Top panels: seasonality obtained using the whole dataset from April 2014 to December 2018; middle panels: seasonality obtained using bi-weekly data collected between August 2016 and December 2018; lower panels: seasonality obtained using dataset filtered for free troposphere from April 2014 to December 2018. Red dashed lines refer to zero variability. The NLS model simulation for time-series gives the detrended seasonality, with error bars indicating one standard deviation of monthly residuals.

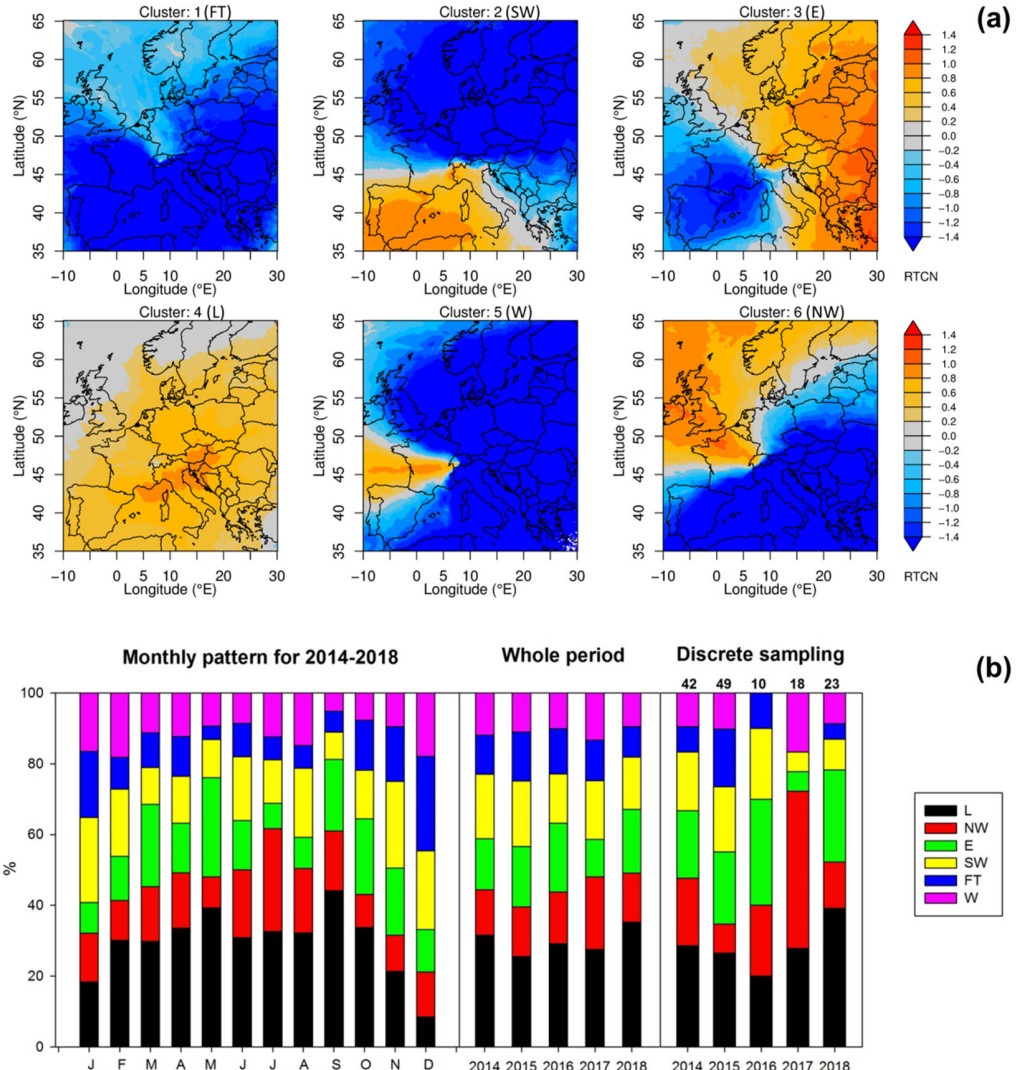

**Figure 4a** Clusters of air mass transport regimes for Jungfraujoch shown as normalized surface source sensitivities over our sampling period. Cluster abbreviations refer to Free Troposphere (FT), Southwest (SW), East (E), Local (L), West (W) and Northwest (NW). The normalization was done by calculating the difference between cluster average source sensitivity and whole period average source sensitivities, divided by the period average. Orange colors indicate the main source regimes in each cluster, whereas blue colors indicate little to no influence on Jungfraujoch observations. The free tropospheric cluster showed lower than average surface sensitivity everywhere.

**4b** Cluster frequency of air mass transport regimes (%) shown as a monthly pattern (left) and interannual patterns for the whole periods (middle) and for the periods of discrete sampling (right). Numbers above the right figure indicate the total number of discrete samples per year.

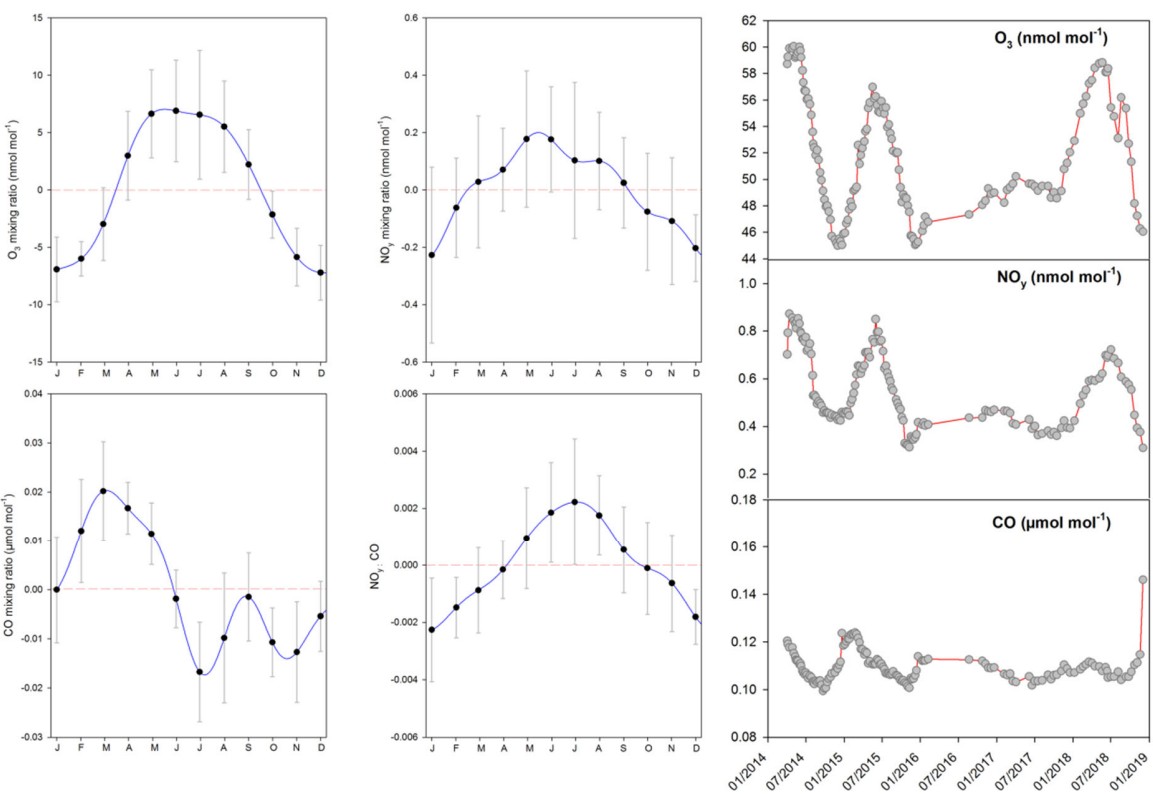

**Figure 5** Left and middle: Seasonality of *in situ* measurements of $O_3$, $NO_y$ and CO mixing ratios and $NO_y$/CO at Jungfraujoch; error bars represent the one standard deviation of monthly residuals from the NLS model simulation for time-series. 10-minute data were used for seasonality analysis.

Right: *In situ* measurements of $O_3$, $NO_y$ and CO mixing ratios averaged over the exact period of discrete sampling (~40 min).

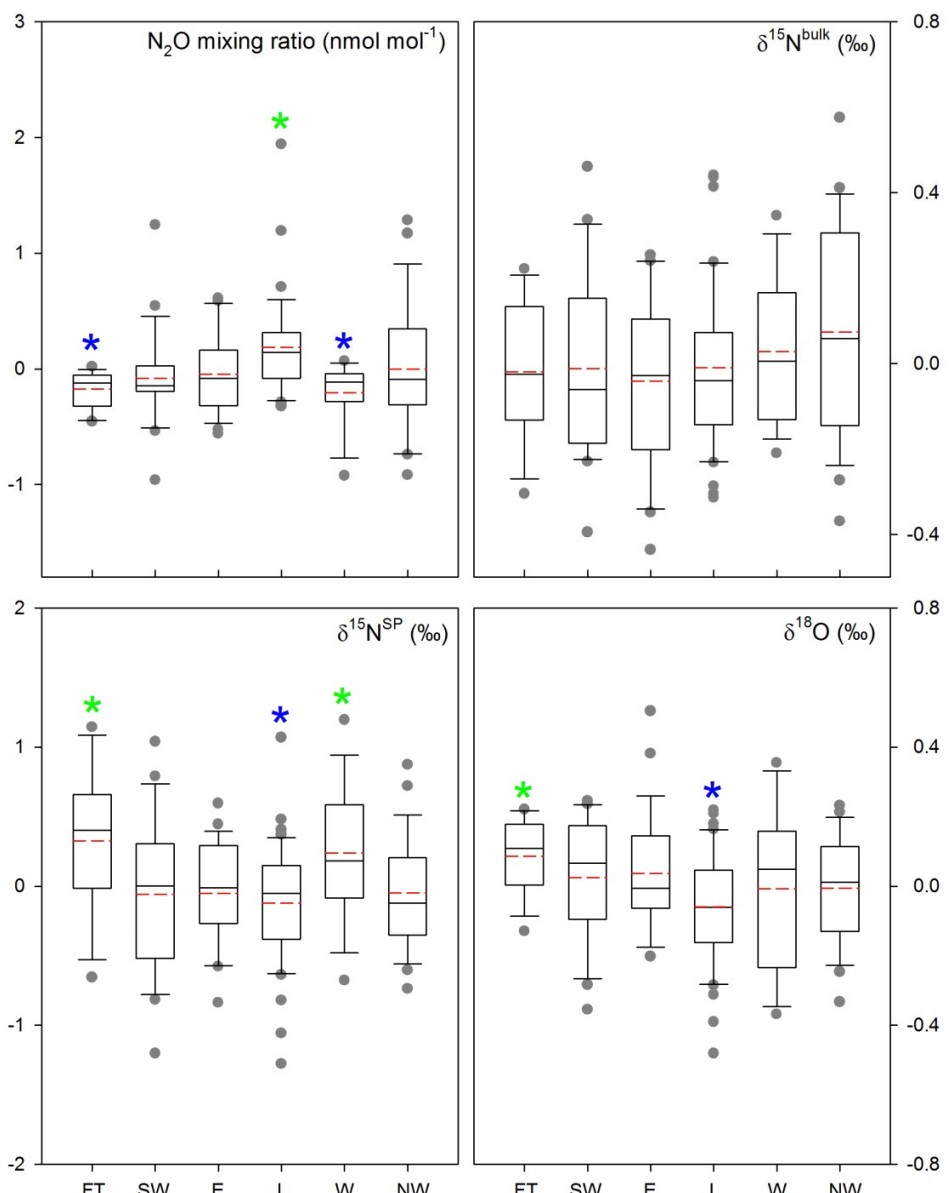

**Figure 6** Comparison of N₂O mixing ratios and isotopic signatures (with linear trends removed) for the six air mass footprint clusters used in the present study. Green and blue stars indicate significantly larger and smaller values than the others, respectively; red dashed lines indicate mean levels; grey points indicate outliers.

1115

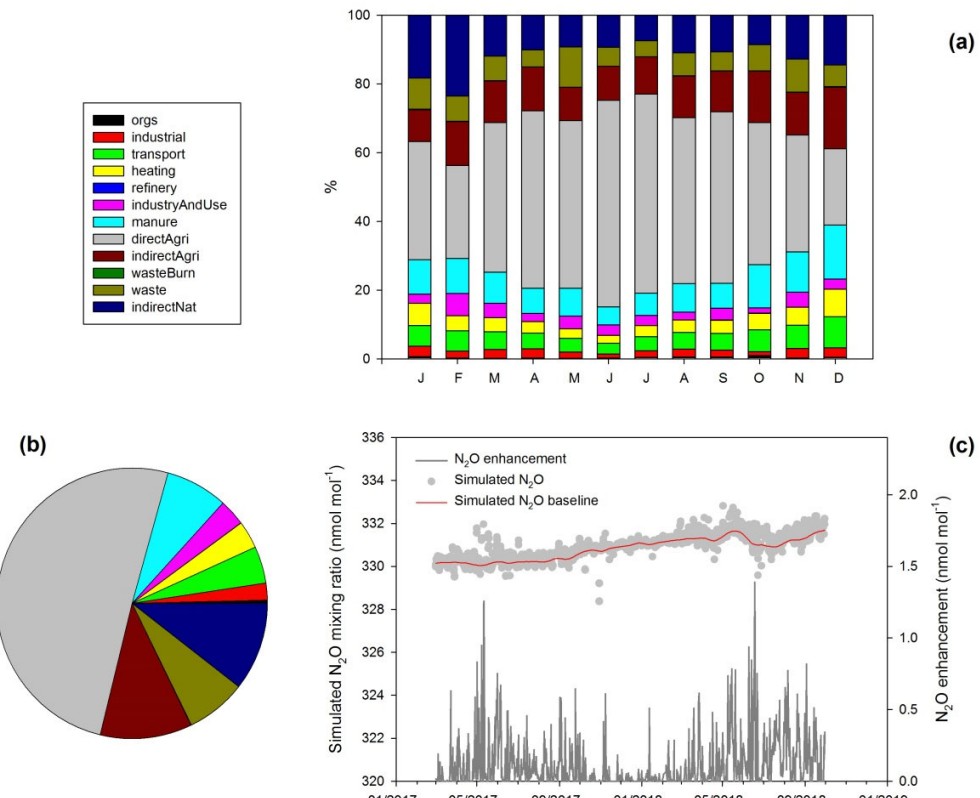

**Figure 7a** Mean monthly stacked-bar plots of source contributions (%) to atmospheric $N_2O$ at Jungfraujoch derived from inversion modeling.

**7b** Overall contributions of $N_2O$ sources responsible for emission to Jungfraujoch.

**7c** Simulated 3-hourly $N_2O$ mixing ratios, $N_2O$ mixing ratio baseline and $N_2O$ enhancements in nmol mol[-1].

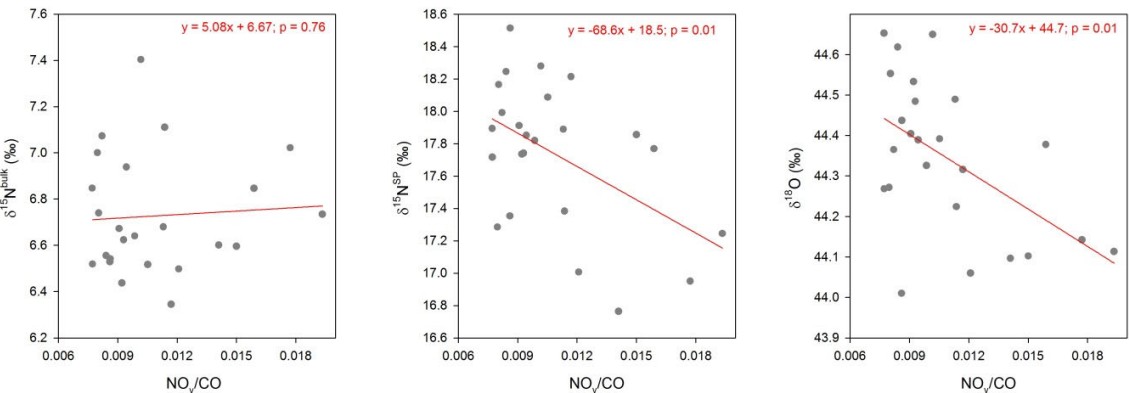

1125    **Figure 8** Relationship between the $NO_y$ to CO ratios and isotopic signatures of $N_2O$; only data points with $NO_y/CO>0.007$ are presented here (which refers to scenarios with strong pollution from local air).

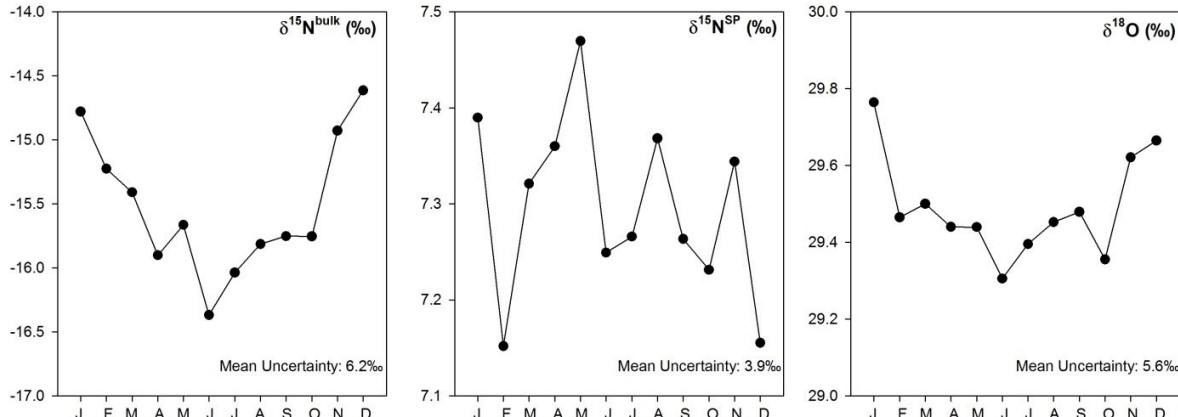

**Figure 9** Simulated seasonal variations of isotopic signatures for overall $N_2O$ sources based on the "bottom-up" approach; uncertainties shown in figures are comparable to the ranges of isotopic signatures for variable sources as found in literature.

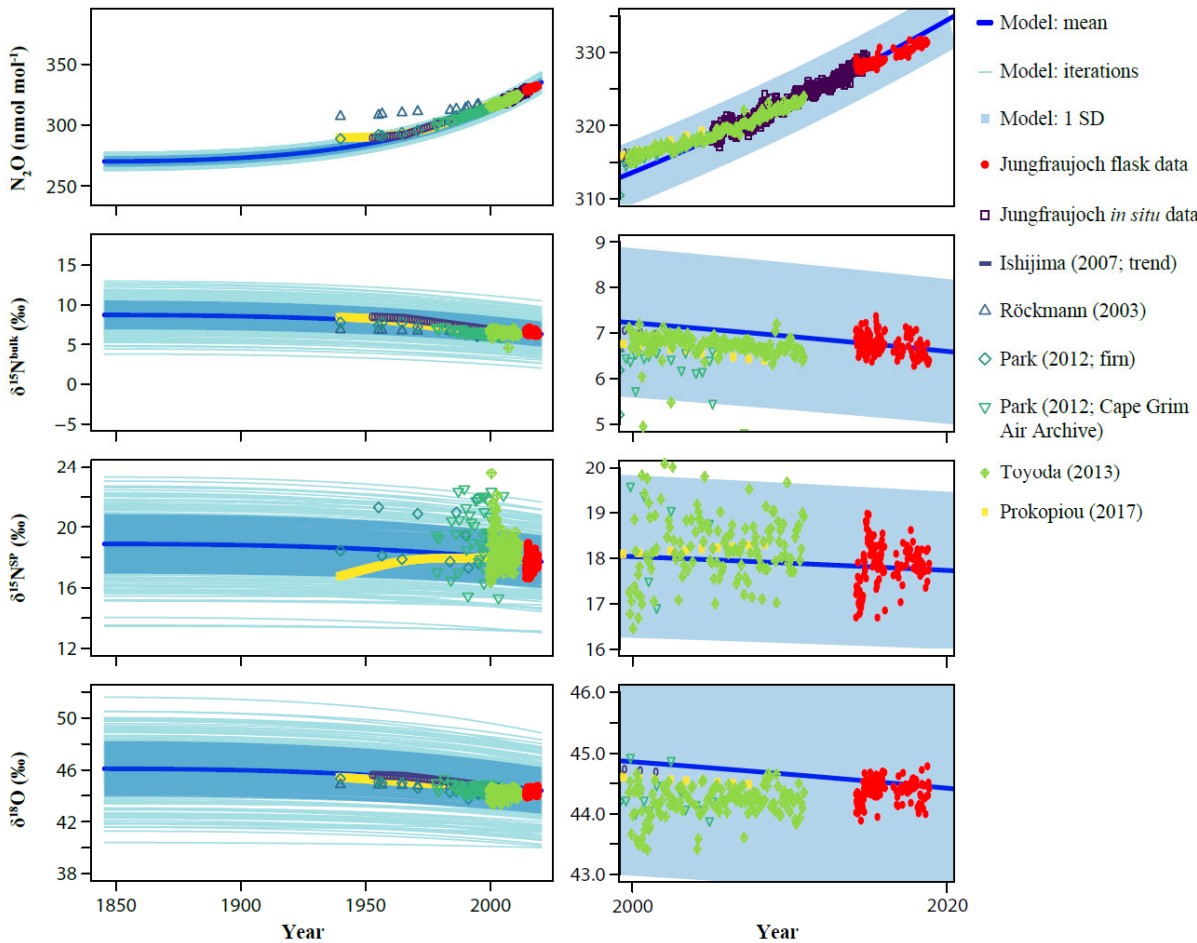

**Figure 10** Two-box model results showing the influence of anthropogenic emissions on $N_2O$ mixing ratio and isotopic composition in the troposphere. Left: full time range from the start of the anthropogenic period (1845) to present day; Right: zoom to the last two decades. Isotopic measurements at Jungfraujoch were used as the only constraint of current tropospheric $N_2O$ isotopic composition for the model. See the materials and method as well as the SI for more details and other input parameters. Atmospheric as well as firn air measurements of $\delta^{15}N^{bulk}$, $\delta^{15}N^{SP}$ and $\delta^{18}O$ from the literature are presented for comparison. Blue shaded areas indicate one standard deviation of the model iterations.