# Peer review of "The isotopic composition of atmospheric nitrous oxide observed at the high-altitude research station Jungfraujoch, Switzerland"

_Atmospheric Chemistry and Physics, 2019_

## Referee Comment (RC1) · Anonymous Referee #1 · 2 Nov 2019

Manuscript: ACP-2019-829 (Yu et al.)  Title: The isotopic composition of atmospheric nitrous oxide observed at the high-altitude research station Jungfraujoch, Switzerland

This manuscript presents measurements of the isotopic composition of N2O obtained from a high-altitude European site – Jungfraujoch in Switzerland, using a recently developed QCLS coupled with a preconcentration unit. The system provided direct and individual measurements of four N2O isotopocules at an ambient level of N2O. From the extensive data sets covering the 5-year study period, authors attempt to derive seasonality and interannual trends in N2O isotopic compositions and discuss them in combination with observed changes in N2O mixing ratio. Overall, the writing and

figures are clear, and the methodology maximizes the functionality of a high-quality dataset. I encourage the publication of this important work, with only a few minor considerations/edits suggested below.

1. LN 186: Sphinx observatory → Sphinx observatory in the Jungfraujoch station

2. LN 357-364: Authors determined annual growth rates of N2O mixing ratio for all in-situ data from 2014 to 2018, with/without the 2014 GC-ECD data, and free tropospheric data only, respectively. Given their 1-sigma values, it seems there are some discrepancies between the entire dataset vs sub-sets of data. Authors did mention some about those discrepancies in lines 548-553. However, if authors thought that they are statistically significant, then additional explanations should be given here, rather than later.

3. LN 375-380: The observed, de-seasonalized trends of delta15N_SP for the whole dataset increased, while delta15N_SP trend showed a decrease when PBL-influenced air samples were excluded. So, authors stated that it implies an impact of local sources. Does it mean that the potential local sources have high delta15N_SP signals? What could it be? Based on the two-box model approach using the current data, authors determined the average isotopic signatures for anthropogenic sources were lower than those for the background troposphere (LN 394-397). If so, the local sources mentioned above could not be associated with anthropogenic sources?

4. LN 405-409: Authors found that there were differences in seasonal patterns of all isotopes between the entire dataset vs. the second phase data. Authors then added that the seasonal variations for free tropospheric samples were similar to those for the whole dataset. Does it imply that the second phase patterns could more represent the PBL-influenced data?

5. LN 421-428: Authors seem to suggest strong exchange with the PBL in summer, based on the observed summer maxima in the monthly seasonal cycles for O3 and NOy mixing ratios. But it is not so clear that the summer maxima in O3 and NOy could

support a stronger air mixing with the surface and thus a PBL impact on the seasonal changes in N2O isotopic compositions, because the maxima in O3 and NOy mixing ratio occur in summer most likely due to stronger sunlight.

6. LN 453-476: In the results section, authors analyzed the seasonal variabilities for not only the entire datasets but also the second phase data, but in the seasonality discussion, the seasonal patterns derived from the second phase data were not discussed, even though the second phase patterns might contains more the surface-influenced signals (see the comment #4). If authors decided not to consider the second phase seasonality, please add statements for the reason in the text.

7. LN 488-505: Fig. 5 demonstrated that direct/indirect agricultural source contributes most to the N2O enhancements, particularly in summer. Then considering peak N2O fluxes and minimum of delta15N_SP observed in Summer, does it suggest that the local agricultural activities enhanced N2O production by "denitrification"? Are there any studies to support this result?

---

## Referee Comment (RC2) · Anonymous Referee #2 · 29 Jan 2020

Review on "**The isotopic composition of atmospheric nitrous oxide observed at the high-altitude research station Jungfraujoch, Switzerland**"

This manuscript described the 5-year observations of nitrous oxide ($N_2O$) mixing ratios and their isotopic compositions at Jungfraujoch using laser spectroscopic technique for the first time. The long-term observations of $N_2O$ isotopocules allow the authors to characterize the integrated isotopic signatures of anthropogenic sources that have been emitted since the industrial revolution and to identify the main processes governing the seasonality of $N_2O$. The authors utilized a two-box model and a Lagrangian particle dispersion model to characterize the isotope signatures of anthropogenic sources that contribute to the atmospheric increase of $N_2O$ concentration. The unique observations of $N_2O$ isotopocules in the middle of the European continent and the interesting interpretation of data makes worth publication. Notwithstanding, there are several hazy spots in the manuscript which needs to be revised in order to avoid any confusion.

Major issues:

1. Application of a two-box model assumes the data obtained at Jungfraujoch to represent the variability of $N_2O$ mixing ratios and its isotopocules in the troposphere. This appear to contradict to the use of footprint model to characterize the isotopic signatures of the anthropogenic sources in the European continent. This is demonstrated in Table 2 and 3 that the isotopic signatures of the anthropogenic $N_2O$ are different. In the text on the lines from 626 to 635, the authors ascribed it to the different isotopic signatures of $N_2O$ source emissions in the model. However, as shown in Table 2 and mentioned in the text (on the line of 612), the single spot observation won't be representative the global scale of atmosphere, but would represent the regional characteristics of $N_2O$. The long-term trends of $N_2O$ isotopocules listed in Table 1 also support that the observation at Jungfraujoch does not represent the tropospheric variability of $N_2O$. Contradict to the global trends of isotopocules shown in Figure 6, the observations of $\delta^{15}N^{SP}$ and $\delta^{18}O$ are positive trends at Jungfraujoch. In view of these contradict aspects revealed in the observation and the model, the isotopic signatures of the anthropogenic source will not help understand the contribution of anthropogenic source to the increase of atmospheric $N_2O$. I would suggest limiting the data interpretation in regional scale.
2. The long-term observation at one station allowed seasonal variation to be explored. The authors argued the minimum $N_2O$ concentration observed in late summer is driven by STE which is also evidenced by the enrichment of $^{15}N$ in the $N_2O$ driven by the photochemical destruction in the stratosphere. On the other hand, $\delta^{15}N^{SP}$ and $\delta^{18}O$ did not seemingly synchronize the STE event, which, the authors argued, the $N_2O$ emitted from the soil overwhelms the effect by STE. If these two processes govern the seasonality of the atmospheric $N_2O$, I would suggest quantifying how to compete these two processes along the year at Jungfraujoch.

Minor issues and technical comments:

1. L 52: The publication year of Tian et al. (2018) is 2019.

2. L 171: "gas chromatography" should be "gas chromatograph" in the context.
3. L 170 – 184: Since no references are given, I suggest describing the analytical methods in detail including the calibration of the system for the analysis of $N_2O$, CO, NOy, and O3 mole fractions perhaps in the section of Supporting Information.
4. L 217: Have you tested the mole fraction dependency of the isotope ratios of $N_2O$? Here, the amount of $N_2O$ for the QCL is 45 ppm. However, Mohn et al. (2010, 2012) concentrated ambient air to > 60 ppm of $N_2O$.
5. L 218: I think the citation of Harris et al. (2017) should be Harris et al. (2014).
6. L 236: What are the matrix gases in CG1 and CG2 standards?
7. L 253: In Figure S2, the scattering of isotope ratios in the second phase look larger than that in the first phase, particularly for $\delta^{18}O$. Is it statistically insignificant?
8. L 313 – 314: $T_{PI}$ and $T_{PD}$ should be replaced to $\tau_{PI}$ and $\tau_{PD}$.
9. L 353: It's misleading. Fig. S3 shows the agreement improved since the year 2015 when GC-ECD was replaced to OA-ICOS, NOT in the second phase.
10. L 358 – 361: Provide the ground that the $N_2O$ growth rates of 0.880±0.001, 0.993±0.001, and 0.93 are in agreement. Statistically they are different each other unless standard deviation of the global growth rate of 0.93 (by NOAA) is larger than ~0.02.
11. L 361: Add the literature (WMO, 2018) next "NOAA (0.93 nmol mol$^{-1}$ a$^{-1}$)".
12. L 362 – 364: The annual growth rate, 0.813±0.027 is not lower than the value 0.858±0.002 within 2 standard deviations.
13. L 376: The authors indicate the insignificant increasing trend of $\delta^{15}N^{SP}$ and $\delta^{18}O$. However, their standard deviations do suggest significant increase of them within 1 sd. It needs to be clarified.
14. L 383 – 391: It needs explanation why the trends of $\delta^{15}N^{SP}$ and $\delta^{18}O$ during the first phase is one order of magnitude larger than that in the second phase.
15. L 438: I would suggest moving Fig. S7 onto the main text as it is the unique visualization to illustrate Lagrangian footprint of isotopic signatures of the sources.
16. L 442 – 451: The section 4.1 does not seem to benefit the main theme of this manuscript. It rather makes the manuscript loose. Analytical quality has already mentioned in the section 2.4 Data analysis (see the lines 246, 252 – 253) and the excellent analytical repeatability for $\delta^{15}N^{SP}$ by QCL is well described in Mohn et al. (2014).
17. L 458: Decock and Six (2013) does not describe the STE process at all. Is it an error in citation?
18. L 459: Add superscript "bulk" next $^{15}N$.
19. L 461: Comparing Figure 3(a) in Toyota et al. (2013) with Figure 1a here, it does not look "almost identical", but perhaps comparable. The monthly mixing ratio of $N_2O$ at Jungfraujoch is at maximum in June while in April at Hateruma Island, Japan.
20. L 464: What are the underlying mechanisms?
21. L 511: Provide the regression coefficients in Figure S8.
22. L 514: $\delta^{15}N^{bulk}$ in Figure S7 is not particularly high in spite of potential influence of STE. It needs to be clarified.
23. L 537: Add minus sigh before 0.06.
24. L 558 – 559: Rahn and Wahlen (2000) do not provide clear evidence on the influence soil water vapor to oxygen isotope in $N_2O$, but they speculated. Thus, it would appropriate to write "… assuming that …" instead of "… given that …".

25. L 605: The authors' argument is not clear here. Based on the isotopic signatures of the anthropogenic $N_2O$, long-term observation at Jungfraujoch indicates the significant contribution of denitrification process in soil while the results from Park et al. (2012) or Prokopiou et al. (2017) favor nitrification process in soil. This is clearly contradicted each other.

26. L 617: Figure 6 shows that $\delta^{15}N^{bulk}$ from Jungfraujoch are higher than any other values including Park et al. (2012) and even Toyota et al. (2013). Thus, this sentence does not help explain why $\delta^{15}N^{bulk}$ of the anthropogenic $N_2O$ from the observation at Jungfraujoch is higher than the value by Park et al. (2012).

27. L 618: It is impossible to mention trends of $\delta^{15}N^{SP}$ as the data is too scattered. In addition, $\delta^{15}N^{SP}$ at Jungfraujoch shows positive trends, too (Table 1).

28. L 652: What do the authors mean the "higher-frequency temporal variation" for $\delta^{15}N^{SP}$ and $\delta^{18}O$? Is it relevant to soil emission? Please state it clearly.

29. L 656: Table 2 clearly shows the isotope signatures from Jungfraujoch differ from the values obtained at other sites, opposite to the statement here.

---

## Author Comment (AC1) · 4 Mar 2020

Referee #1

Manuscript: ACP-2019-829 (Yu et al.) Title: The isotopic composition of atmospheric nitrous oxide observed at the high-altitude research station Jungfraujoch, Switzerland. This manuscript presents measurements of the isotopic composition of $N_2O$ obtained from a high-altitude European site – Jungfraujoch in Switzerland, using a recently developed QCLS coupled with a preconcentration unit. The system provided direct and individual measurements of four $N_2O$ isotopocules at an ambient level of $N_2O$. From the extensive data sets covering the 5-year study period, authors attempt to derive seasonality and interannual trends in $N_2O$ isotopic compositions and discuss them in combination with observed changes in $N_2O$ mixing ratio. Overall, the writing and figures are clear, and the methodology maximizes the functionality of a high-quality dataset. I encourage the publication of this important work, with only a few minor considerations/edits suggested below.

1. LN 186: Sphinx observatory→ Sphinx observatory in the Jungfraujoch station

R: OK

2. LN 357-364: Authors determined annual growth rates of $N_2O$ mixing ratio for all in-situ data from 2014 to 2018, with/without the 2014 GC-ECD data, and free tropospheric data only, respectively. Given their 1-sigma values, it seems there are some discrepancies between the entire dataset vs sub-sets of data. Authors did mention some about those discrepancies in lines 548-553. However, if authors thought that they are statistically significant, then additional explanations should be given here, rather than later.

R: Thanks for the suggestion. In section 3.1, we have further elaborated this: "This difference in $N_2O$ growth rates is probably due to the limited data quality of GC-ECD, although a lower growth rate in 2014 compared to 2015-2018 cannot be excluded.".

3. LN 375-380: The observed, de-seasonalized trends of delta15N_SP for the whole dataset increased, while delta15N_SP trend showed a decrease when PBL-influenced air samples were excluded. So, authors stated that it implies an impact of local sources. Does it mean that the potential local sources have high delta15N_SP signals? What could it be? Based on the two-box model approach using the current data, authors determined the average isotopic signatures for anthropogenic sources were lower than those for the background troposphere (LN 394-397). If so, the local sources mentioned above could not be associated with anthropogenic sources?

R: The authors agree, that the increasing trend of $d^{15}N^{SP}$ observed between April 2014 and December 2018 at Jungfraujoch (Fig. 2) and the decreasing trend over longer timescales as derived with a two-box modelling approach using the EDGAR emission inventory (Fig. 6; original version) might look inconsistent.

However, it is noteworthy that, the deseasonalized trends of $\delta^{15}N^{SP}$ at Jungfraujoch were not statistically significant, with/without filtering for impact from planetary boundary layer (LN 375-377_Original version). The only significantly positive trend

of $\delta^{15}N^{SP}$ was found in the first phase (Table 1). Although mean $\delta^{15}N^{SP}$ values of $N_2O$ sources according to EDGAR emission inventory are lower than that observed in tropospheric background (Table S2), a changing proportion of $N_2O$-emitting soil process, i.e. nitrification vs. denitrification, with $\delta^{15}N^{SP}$ values of 33‰, as compared to about 0‰ (Sutka et al., 2006), might rationalize this inconsistency. This shift in the isotopic signatures of anthropogenic sources, might be interpreted as a climate change feedback, as discussed in section 4.4. Similarly, Park et al. (2012) attributed an increase rate of 0.06‰ $a^{-1}$ in $\delta^{15}N^{SP}$ in 2005 to 10% increase in the relative contribution of nitrification to global $N_2O$ production since 1975. This is already discussed in section 4.3.

In the two-box model approach, the estimation of isotopic signatures for anthropogenic sources mainly depends on the measured current and predefined preindustrial $N_2O$ mixing ratios and isotopic signatures (Table S1). As shown in Figure 6 (original version), the simulated trend of $\delta^{15}N^{SP}$ in the troposphere is negative, consistent with the lower $\delta^{15}N^{SP}$ for anthropogenic sources than for the tropospheric background. The current (insignificant) increase in $d^{15}N^{SP}$ at Jungfraujoch, might be evaluated with the two-box model approach in the future, if extended time-series of isotope data will become available (e.g. Prokopiou et al., 2017).

4. LN 405-409: Authors found that there were differences in seasonal patterns of all isotopes between the entire dataset vs. the second phase data. Authors then added that the seasonal variations for free tropospheric samples were similar to those for the whole dataset. Does it imply that the second phase patterns could more represent the PBL-influenced data?

R: As indicated in LN 401-409 (original version) and Figure 3, we found a significant seasonal pattern of $\delta^{15}N^{SP}$, with a summer minimum, for both the whole dataset and the second phase. For $\delta^{15}N^{bulk}$, a significant seasonal pattern was seen in the whole dataset but not in the second phase (seasonal variability > uncertainty). Hence, our results do not necessarily indicate that seasonal patterns were different between the entire and the second-phase data.

Air mass footprints suggest that, in 2017 (second phase), discrete sampling received less contribution from free troposphere than in the other years (Fig. 4b), possibly pointing to a stronger influence of PBL. However, this is not supported by *in situ* $NO_y$ and CO measurements (Figure S6b; no clear difference), which has been suggested as a more effective indicator for free troposphere (Herrmann et al., 2015). Given the larger uncertainty in seasonality-analysis due to lower sampling frequency in the second phase (Section 3.4), it is difficult to draw the conclusion that such "insignificant" changes in seasonal patterns in the second phase are due to a stronger PBL influence.

5. LN 421-428: Authors seem to suggest strong exchange with the PBL in summer, based on the observed summer maxima in the monthly seasonal cycles for $O_3$ and $NO_y$ mixing ratios. But it is not so clear that the summer maxima in $O_3$ and $NO_y$ could support a stronger air mixing with the surface and thus a PBL impact on the seasonal

changes in $N_2O$ isotopic compositions, because the maxima in $O_3$ and $NO_y$ mixing ratio occur in summer most likely due to stronger sunlight.

R: We agree that $O_3$ alone may not be a good indicator for air exchange with PBL, as elevated $O_3$ concentration at Jungfraujoch can be due to air exchange with PBL and/or stratosphere. Therefore, the text in section 3.4 has now been revised. However, $NO_y$ : CO used in this study has been previously tested to be an effective indicator for determining the age of air mass, i.e. to identify recently polluted air transported to Jungfraujoch from the PBL (Herrmann et al., 2015; Zellweger et al., 2003). In addition, air mass footprint analysis supports such pattern with lowest source sensitivity from free troposphere in summer (Figure 4).

6. LN 453-476: In the results section, authors analyzed the seasonal variabilities for not only the entire datasets but also the second phase data, but in the seasonality discussion, the seasonal patterns derived from the second phase data were not discussed, even though the second phase patterns might contains more the surface-influenced signals (see the comment #4). If authors decided not to consider the second phase seasonality, please add statements for the reason in the text.

R: For the second phase, the seasonal patterns of $\delta^{15}N^{bulk}$ and $\delta^{18}O$ were not significant, while $\delta^{15}N^{SP}$ showed a significant seasonal pattern similar to that for the whole dataset. Therefore, it was not specifically discussed. Nonetheless, we thank the reviewer for the suggestion and have added a few more discussion points in section 4.2, regarding the seasonal variabilities of $N_2O$ isotopic signature in the second phase.

7. LN 488-505: Fig. 5 demonstrated that direct/indirect agricultural source contributes most to the $N_2O$ enhancements, particularly in summer. Then considering peak $N_2O$ fluxes and minimum of delta15N_SP observed in summer, does it suggest that the local agricultural activities enhanced $N_2O$ production by "denitrification"? Are there any studies to support this result?

R: Yes, one isotopic study of $N_2O$ emissions from Swiss grassland (Wolf et al., 2015) suggested that $N_2O$ emissions in summer periods were mostly contributed by denitrification, given that high $N_2O$ fluxes were associated with low $\delta^{15}N^{SP}$ values (below 5‰). This has been confirmed again in a recent study (Ibraim et al., submitted to Global Biogeochemical Cycles) showing that the $\delta^{15}N^{SP}$ of $N_2O$ emitted from a managed grassland during a late summer was consistently within a small range of 0-10‰, regardless of soil water-filled-pore-space.

**Reference**

Herrmann, E., Weingartner, E., Henne, S., Vuilleumier, L., Bukowiecki, N., Steinbacher, M., Conen, F., Coen, M. C., Hammer, E., Juranyi, Z., Baltensperger, U. and Gysel, M.: Analysis of long-term aerosol size distribution data from Jungfraujoch with emphasis on free tropospheric conditions, cloud influence, and air mass transport, J. Geophys. Res. Atmos., 120, 1751–1762, doi:10.1002/2014JD022963.Received, 2015.

Prokopiou, M., Martinerie, P., Sapart, C. J., Witrant, E., Monteil, G. A., Ishijima, K., Bernard, S., Kaiser, J., Levin, I., Sowers, T., Blunier, T., Etheridge, D., Dlugokencky, E., van de Wal, R. S. W. and Röckmann, T.: Constraining $N_2O$ emissions since 1940 using firn air isotope measurements in both hemispheres, Atmos. Chem. Phys., 2011(June), 1–50, doi:10.5194/acp-2016-487, 2017.

Sutka, R. L., Ostrom, N. E., Ostrom, P. H., Breznak, J. A, Pitt, A. J., Li, F. and Gandhi, H.: Distinguishing Nitrous Oxide Production from Nitrification and Denitrification on the Basis of Isotopomer Abundances Distinguishing Nitrous Oxide Production from Nitrification and Denitrification on the Basis of Isotopomer Abundances, Appl. Environ. Microbiol., 72(1), 638–644, doi:10.1128/AEM.72.1.638, 2006.

Wolf, B., Merbold, L., Decock, C., Tuzson, B., Harris, E., Six, J., Emmenegger, L. and Mohn, J.: First on-line isotopic characterization of $N_2O$ above intensively managed grassland, Biogeosciences, 12(8), 2517–2531, doi:10.5194/bg-12-2517-2015, 2015.

Zellweger, C., Forrer, J., Hofer, P., Nyeki, S., Schwarzenbach, B., Weingartner, E., Ammann, M. and Baltensperger, U.: Partitioning of reactive nitrogen ($NO_y$) and dependence on meteorological conditions in the lower free troposphere, Atmos. Chem. Phys., 3(3), 779–796, doi:10.5194/acp-3-779-2003, 2003.

---

## Author Comment (AC2) · 4 Mar 2020

Referee #2
Review on "**The isotopic composition of atmospheric nitrous oxide observed at the high-altitude research station Jungfraujoch, Switzerland**".

This manuscript described the 5-year observations of nitrous oxide ($N_2O$) mixing ratios and their isotopic compositions at Jungfraujoch using laser spectroscopic technique for the first time. The long-term observations of $N_2O$ isotopocules allow the authors to characterize the integrated isotopic signatures of anthropogenic sources that have been emitted since the industrial revolution and to identify the main processes governing the seasonality of $N_2O$. The authors utilized a two-box model and a Lagrangian particle dispersion model to characterize the isotope signatures of anthropogenic sources that contribute to the atmospheric increase of $N_2O$ concentration. The unique observations of $N_2O$ isotopocules in the middle of the European continent and the interesting interpretation of data makes worth publication. Notwithstanding, there are several hazy spots in the manuscript which needs to be revised in order to avoid any confusion.

Major issues:
1. Application of a two-box model assumes the data obtained at Jungfraujoch to represent the variability of $N_2O$ mixing ratios and its isotopocules in the troposphere. This appear to contradict to the use of footprint model to characterize the isotopic signatures of the anthropogenic sources in the European continent. This is demonstrated in Table 2 and 3 that the isotopic signatures of the anthropogenic $N_2O$ are different. In the text on the lines from 626 to 635, the authors ascribed it to the different isotopic signatures of $N_2O$ source emissions in the model. However, as shown in Table 2 and mentioned in the text (on the line of 612), the single spot observation won't be representative the global scale of atmosphere, but would represent the regional characteristics of $N_2O$. The long-term trends of $N_2O$ isotopocules listed in Table 1 also support that the observation at Jungfraujoch does not represent the tropospheric variability of $N_2O$. Contradict to the global trends of isotopocules shown in Figure 6, the observations of $\delta^{15}N^{SP}$ and $\delta^{18}O$ are positive trends at Jungfraujoch. In view of these contradict aspects revealed in the observation and the model, the isotopic signatures of the anthropogenic source will not help understand the contribution of anthropogenic source to the increase of atmospheric $N_2O$. I would suggest limiting the data interpretation in regional scale.
R: We thank the review for the critics and suggestions. Although the reviewer suggests to limit our data interpretation with respect to the global model, we argue that the air samples collected from Jungfraujoch Sphinx still represent the background troposphere, despite the contribution of regional emissions to the seasonal variability. The box model estimates for the emission strength and isotopic composition of the anthropogenic source are largely depending on the mean values of $N_2O$ concentrations and isotopic composition at Jungfraujoch, and little affected by subtle temporal changes, which are shown in the seasonal variabilities. Based on the $NO_y$ : CO criterion (Herrmann et al., 2015; Zellweger et al., 2003), which has been identified as an effective indicator for the (short) age of air mass, 110 out of 142 sample points were found to represent the free troposphere. To demonstrate that two-box model results are not affected by regional emissions, we re-ran the two-box model with the data filtered for free troposphere and got statistically identical results. The new results are now mentioned in section 4.4.
Regarding the reviewer's arguments referring to Table 2 and 3 as well as the texts in discussion, we believe that there are misinterpretations. In our discussion (LN 626-635), the differences in source isotopic signatures between Table 2 (two-box model) and Table 3 (bottom-up estimate) was largely attributed to the uncertainty in the estimated source isotopic signatures, which were used in the bottom-up model (Table S2; original version). This was further explained by comparing our bottom-up estimates with those from Toyoda et al. (2013), demonstrating that the selection of source isotopic signatures for distinct source categories from literature largely influence the isotopic composition of the anthropogenic source.

We are aware that a single-site study can be limited in determining long-term trends of $N_2O$ isotopic signatures. As we discussed in the manuscript, extension of the study period at an even higher sampling frequency would reduce such uncertainties. Although the interannual trends of $\delta^{15}N^{SP}$ and $\delta^{18}O$ were positive in the first phase of our observation, we obtained insignificant trends for the whole dataset, which in return makes a minor influence on the model estimates. Given the relatively short study period, the mean isotopic signatures observed at Jungfraujoch is more important than the trends for determining isotopic signatures of anthropogenic sources from the NH background atmosphere.

2. The long-term observation at one station allowed seasonal variation to be explored. The authors argued the minimum $N_2O$ concentration observed in late summer is driven by STE which is also evidenced by the enrichment of $^{15}N$ in the $N_2O$ driven by the photochemical destruction in the stratosphere. On the other hand, $\delta^{15}N^{SP}$ and $\delta^{18}O$ did not seemingly synchronize the STE event, which, the authors argued, the $N_2O$ emitted from the soil overwhelms the effect by STE. If these two processes govern the seasonality of the atmospheric $N_2O$, I would suggest quantifying how to compete these two processes along the year at Jungfraujoch.

R: The authors agree, as already mentioned in the manuscript, that $N_2O$ isotopic composition at Jungfraujoch is controlled by stratospheric intrusions and uplift of polluted air masses. However, it is currently not possible to quantify the relative importance of these two mechanisms over time, given that temporally resolved isotopic signatures of stratospheric air and soil $N_2O$ sources are not available for Jungfraujoch. We simulated the contribution of upper tropospheric air (15 km) to Jungfraujoch station, which is highest in the August. This acts as a qualitative indicator of the seasonal pattern of STE, which assists to explain the seasonal variability of $\delta^{15}N^{bulk}$ (added to the discussion 4.2). On the other hand, simulations of $N_2O$ enhancements (on average 60% from soil) for 2017-2018 suggest that ground emissions of $N_2O$ were highest in the early to middle summer (Fig. 5; original version). In August, when $N_2O$ mixing ratios were lowest below baseline, the $N_2O$ depletion due to mixing with stratospheric air clearly outcompeted the enhancement from ground emissions.

In a back-of-the-envelope calculation, we assume $N_2O$ enhancement from ground-based emissions in August to be 0.15-0.20 nmol mol$^{-1}$, which is close with or slightly smaller than the maximum change of $N_2O$ mixing ratio above baseline (0.20 nmol mol$^{-1}$; Fig. 1b). Then, given that the net minimum of $N_2O$ mixing ratio in August is -0.2 nmol mol$^{-1}$ below baseline, we can estimate the $N_2O$ depletion due to STE as 0.35-0.40 nmol mol$^{-1}$. In addition, $N_2O$ enhancement by soil emission (60% of total ground emission) can be calculated as 0.09-0.12 nmol mol$^{-1}$. With the isotopic effect associated with each mechanism from literature, we may estimate the combined effects of the two mechanisms on the maximum variabilities of $\delta^{15}N^{SP}$ in the late summer at Jungfraujoch. The net isotopic effect of mixing with stratospheric air is assumed to be about +5‰ for the lower stratosphere (higher isotopic signature but smaller mixing ratio for higher stratosphere) (Toyoda et al., 2018); the isotopic effect due to switch from nitrification to denitrification is assumed to be -30‰ (Sutka et al., 2006). Therefore, STE contributes $N_2O$ depletion at a strength four times of that from soil emissions, while the isotopic effect of STE is only 1/6. Based on the estimates above, it is reasonable to suggest that soil emission would outcompete STE in regulating $\delta^{15}N^{SP}$ during the late summer. Nevertheless, our estimates may have large uncertainty, and require further validation with isotopic measurements of two individual processes. By contrast, given that the isotopic effects of soil processes are much smaller for $\delta^{15}N^{bulk}$ (Toyoda et al., 2011), STE stands out to control the variability of $\delta^{15}N^{bulk}$ during late summer. We have now implemented these estimates in the supplementary material and have included more discussion in the manuscript.

02
Minor issues and technical comments:
1. L 52: The publication year of Tian et al. (2018) is 2019.
2. L 171: "gas chromatography" should be "gas chromatograph" in the context.
R: OK.

3. L 170 – 184: Since no references are given, I suggest describing the analytical methods in detail including the calibration of the system for the analysis of $N_2O$, CO, $NO_y$, and $O_3$ mole fractions perhaps in the section of Supporting Information.
R: Thanks for the suggestion. Additional details on the analytical method of $N_2O$ is now implemented. The references for CO, $NO_y$ and $O_3$ were given in the section 2.1 for atmospheric pollutant measurements at Jungfraujoch. In the revision, we have referred to specific publications for each pollutant giving more details on analytical methods.

4. L 217: Have you tested the mole fraction dependency of the isotope ratios of $N_2O$? Here, the amount of $N_2O$ for the QCL is 45 ppm. However, Mohn et al. (2010, 2012) concentrated ambient air to > 60 ppm of $N_2O$.
R: Yes. The dependency of $N_2O$ mole fraction on isotopic results was determined and corrected for (if necessary) during every batch of measurement. In addition, following identical-treatment principle, we fixed the $N_2O$ mole fractions of calibration standards (CG1 and CG2) to the same level 45 ppm.

5. L 218: I think the citation of Harris et al. (2017) should be Harris et al. (2014).
R: Not true. Harris et al. (2014) described the laser spectroscopic technique that was developed in MIT for $N_2O$ isotopic measurement; however, this study shares the same instrumentation as Harris et al. (2017) which was developed at Empa (Switzerland).

6. L 236: What are the matrix gases in CG1 and CG2 standards?
R: 78% $N_2$ and 21% $O_2$. This is now mentioned in the manuscript.

7. L 253: In Figure S2, the scattering of isotope ratios in the second phase look larger than that in the first phase, particularly for $\delta^{18}O$. Is it statistically insignificant?
R: Statistically, the difference is not significant.

8. L 313 – 314: $T_{PI}$ and $T_{PD}$ should be replaced to $\tau_{PI}$ and $\tau_{PD}$.
R: OK

9. L 353: It's misleading. Fig. S3 shows the agreement improved since the year 2015 when GC-ECD was replaced to OA-ICOS, NOT in the second phase.
R: This is now revised. See section 3.1 for change.

10. L 358 – 361: Provide the ground that the $N_2O$ growth rates of 0.880±0.001, 0.993±0.001, and 0.93 are in agreement. Statistically they are different each other unless standard deviation of the global growth rate of 0.93 (by NOAA) is larger than ~0.02.
R: We agree. The uncertainty of growth rates by NOAA is around 0.03 nmol mol$^{-1}$ a$^{-1}$, suggesting that the global mean growth rate of 0.93 ± 0.03 nmol mol$^{-1}$ a$^{-1}$ is lower than retrieved from our measurements at Jungfraujoch, excluding GC-ECD measurements (2015-2018). This is now revised.

11. L 361: Add the literature (WMO, 2018) next "NOAA (0.93 nmol mol$^{-1}$ a$^{-1}$)".
R: OK.

12. L 362 – 364: The annual growth rate, 0.813±0.027 is not lower than the value 0.858±0.002 within 2 standard deviations.
R: This is now revised as "the absolute growth rate determined from the discrete gas samples was even lower albeit larger uncertainty ($0.813 \pm 0.027$ nmol mol$^{-1}$ a$^{-1}$)".

13. L 376: The authors indicate the insignificant increasing trend of $\delta^{15}N^{SP}$ and $\delta^{18}O$. However, their standard deviations do suggest significant increase of them within 1 sd. It needs to be clarified.
R: In Table 1, we showed coefficients from linear regressions with 1 SD. However, as indicated in section 2.6, significance level for linear regression was set to $p < 0.01$ (confidence level of 99%). Hence, this would require coefficients to be larger than 3 times of SD.

14. L 383 – 391: It needs explanation why the trends of $\delta^{15}N^{SP}$ and $\delta^{18}O$ during the first phase is one order of magnitude larger than that in the second phase.
R: As stated in LN 386-388 (original version), the strong increasing trends for $\delta^{15}N^{SP}$ and $\delta^{18}O$ were most likely due to the unexpectedly low $\delta^{15}N^{SP}$ and $\delta^{18}O$ values in summer 2014 (Fig. 2). In addition, this has been discussed in section 4.3: "Nevertheless, our observation period is shorter than that of other studies, so the interannual trends determined here are more likely affected by year-to-year variability" (LN 540-542; original version).

15. L 438: I would suggest moving Fig. S7 onto the main text as it is the unique visualization to illustrate Lagrangian footprint of isotopic signatures of the sources.
R: Agree.

16. L 442 – 451: The section 4.1 does not seem to benefit the main theme of this manuscript. It rather makes the manuscript loose. Analytical quality has already mentioned in the section 2.4 Data analysis (see the lines 246, 252 – 253) and the excellent analytical repeatability for $\delta^{15}N^{SP}$ by QCL is well described in Mohn et al. (2014).
R: Although an excellent repeatability of singular measurements has been shown by (Mohn et al., 2014), it is important that repeated measurements of target gases show a good consistency, indicating long-term robustness of our measurements. This is crucial for isotopic measurements of background atmosphere, as target variabilities of our samples are most likely in a range that is only a few times larger than our analytical precision (Toyoda et al., 2013). Therefore, we would like to keep this section. To avoid confusion, we have now changed "analytical repeata-bility" to "target repeatability".

17. L 458: Decock and Six (2013) does not describe the STE process at all. Is it an error in citation?
R: The reviewer is right. We have now revised the citation.

18. L 459: Add superscript "bulk" next 15N.
R: Superscript "bulk" is used for $\delta^{15}N$ values, which refer to the average of $\delta^{15}N^{\alpha}$ and $\delta^{15}N^{\beta}$. Here, enrichment of $^{15}N$ is a general description, thus not requiring "bulk" notation.

19. L 461: Comparing Figure 3(a) in Toyota et al. (2013) with Figure 1a here, it does not look "almost identical", but perhaps comparable. The monthly mixing ratio of $N_2O$ at Jungfraujoch is at maximum in June while in April at Hateruma Island, Japan.
20. L 464: What are the underlying mechanisms?

21. L 511: Provide the regression coefficients in Figure S8.
R: They have been already embedded in each figure as red fonts.

22. L 514: $\delta^{15}N^{bulk}$ in Figure S7 is not particularly high in spite of potential influence of STE. It needs to be clarified.
R: This must be a misunderstanding. In Figure S7, we compared $\delta^{15}N^{bulk}$ for six air mass footprint clusters but not showing air coming from stratosphere.

23. L 537: Add minus sigh before 0.06.
24. L 558 – 559: Rahn and Wahlen (2000) do not provide clear evidence on the influence soil water vapor to oxygen isotope in $N_2O$, but they speculated. Thus, it would appropriate to write "… assuming that …" instead of "… given that …".
R: OK

25. L 605: The authors' argument is not clear here. Based on the isotopic signatures of the anthropogenic $N_2O$, long-term observation at Jungfraujoch indicates the significant contribution of denitrification process in soil while the results from Park et al. (2012) or Prokopiou et al. (2017) favor nitrification process in soil. This is clearly contradicted each other.
R: Based on the difference between our and other studies in box-model estimates, we suggest that the isotopic signatures of anthropogenic sources may have shifted in recent decades. This would mean non-linear change of $N_2O$ source isotopic signatures since preindustrial times. On the other hand, the uncertainty in measuring $N_2O$ isotopic signatures in the background atmosphere and inter-comparability among laboratories may play a role in the discrepancy of the estimated source isotopic signatures. Further elaborations are incorporated now (section 4.4).

26. L 617: Figure 6 shows that $\delta^{15}N^{bulk}$ from Jungfraujoch are higher than any other values including Park et al. (2012) and even Toyota et al. (2013). Thus, this sentence does not help explain why $\delta^{15}N^{bulk}$ of the anthropogenic $N_2O$ from the observation at Jungfraujoch is higher than the value by Park et al. (2012).
27. L 618: It is impossible to mention trends of $\delta^{15}N^{SP}$ as the data is too scattered. In addition, $\delta^{15}N^{SP}$ at Jungfraujoch shows positive trends, too (Table 1).
R: Thank you for the critical comments. We have now clarified these two points in the discussion. Below are some explanations.
The difference in $\delta^{15}N^{bulk}$ between our study and Toyoda et al. (2013) is relatively small (0.10-0.15‰ based on year-to-year comparison) compared with the difference between ours and Park et al. (2012) (0.40-0.5‰). Therefore, the $\delta^{15}N^{bulk}$ of anthropogenic source estimated with two-box model is much smaller in Park et al. (2012) than in ours and Toyoda et al. (2013). Even larger inter-laboratory differences in $\delta^{15}N^{bulk}$ have been observed in Ostrom et al. (2018) and can be explained by different anchoring to international scales (Air-N2).
As stated in section 4.4, the difference between current mean tropospheric isotopic values and preindustrial values (given in Table S1) are important in determining the trend of $N_2O$ isotopic signatures in the model estimates. The trends mentioned here are referred to long-term trends since preindustrial times as simulated by the model, but not the observed trends in the "current" troposphere.

28. L 652: What do the authors mean the "higher-frequency temporal variation" for $\delta^{15}N^{SP}$ and $\delta^{18}O$? Is it relevant to soil emission? Please state it clearly.
R: This is not referred to soil emissions. The determined interannual trends for $\delta^{15}N^{SP}$ and $\delta^{18}O$ showed large uncertainties, which is possibly due to large temporal (seasonal) variabilities of $\delta^{15}N^{SP}$ and $\delta^{18}O$. We have reformulated this statement.

29. L 656: Table 2 clearly shows the isotope signatures from Jungfraujoch differ from the values obtained at other sites, opposite to the statement here.

R: Within model uncertainty, our model estimates of isotopic signatures for anthropogenic sources were largely in agreement with the other studies, except for the $\delta^{15}N^{bulk}$ and $\delta^{15}N^{SP}$ when compared with Park et al. (2012) and Prokopiou et al. (2017).

**Reference**

[revised manuscript text omitted]

---

## Editor Decision (ED1)

**Manuscript structure:** To avoid fragmentation and for ease of readability, tables S1 and S2 as well as Figures S3 to S9 should be moved to the main part of the paper. They are key parts of the Results and Discussion sections and in many cases referred to repeatedly in the text. Since Table S3 only adds one additional column with new values compared with Table 2, it should be merged with Table 2.

**Data availability:** Please upload all data used for the study to a publicly accessible archive and provide a Digital Object Identifier (DOI). It is not sufficient to have them available on request from the lead author. For details, please refer to [https://www.atmospheric-chemistry-and-physics.net/about/data_policy.html](https://www.atmospheric-chemistry-and-physics.net/about/data_policy.html)

**Mixing calculation:** There is an error in your "Estimates of the combined effects from STE and soil emissions on $\delta^{15}N^{SP}$". The effect on $\delta$ due to mixing with stratospheric air is about three times larger than the 5 ‰ value you used. In contrast, the effect due to soil emission is too big and should be about three times smaller. However, overall, the combined effect is too small to be measured.

I think this is an important calculation and you should move it from the supplement to the main text (after correcting it as explained in the following).

*Mixing with stratospheric air*: The effect on tropospheric air can be approximated using the apparent isotopic fractionation $\varepsilon_{app}$ (Kaiser et al. 2006). You can readily derive this from the slope of the Rayleigh fractionation equation at $y = y_T$ (where $y_T$ is the N$_2$O mole fraction at the point the of mixing between troposphere and stratosphere):

$$\delta = (1+\delta_0)f^{\varepsilon_{app}} - 1 \quad \text{with } f = \frac{y}{y_T}$$

$$\frac{\mathrm{d}\delta}{\mathrm{d}f} = (1+\delta_0)\varepsilon_{app}f^{\varepsilon_{app}-1}$$

$$y_T\frac{\mathrm{d}\delta}{\mathrm{d}y} = (1+\delta_0)\varepsilon_{app}\left(\frac{y}{y_T}\right)^{\varepsilon_{app}-1}$$

This gives

$$\lim_{y\to y_T}\frac{\mathrm{d}\delta}{\mathrm{d}y} = \frac{(1+\delta_0)\varepsilon_{app}}{y_T}$$

The derived slope is in agreement with the compact relationship between $\delta$ and N$_2$O mole fraction plotted in Fig. 5 of Kaiser et al. (2006).

For the "site-preference" $\delta^{15}N^{SP}$, $\varepsilon_{app}$ needs to be replaced with the difference between the $^{15}N/^{14}N$ isotope fractionations at the central and terminal N atoms, $^2\varepsilon_{app}-^1\varepsilon_{app}$ (or $^\alpha\varepsilon_{app}-^\beta\varepsilon_{app}$).

For the lower stratosphere, $\varepsilon_{app}(^{15}N^{SP}) \approx -15$ ‰ (Kaiser et al. 2006), i.e. about three times larger than your value of 5 ‰.

*Effect of soil emissions:* I am not sure what you mean by the sentence "The isotopic effect of soil emission (–30‰), which is mainly attributed to the switch from nitrification to denitrification, is taken from Sutka et al. (2006)." It sounds as if you have taken the difference between the $\delta^{15}N^{SP}$

values of denitrification (0 ‰?) and nitrification (+30 ‰?). However, the relevant quantity here is the difference between the $\delta^{15}N^{SP}$ values of soil emissions (+7.3 ‰; Table 3) and tropospheric air (+18 ‰; Fig. 2), i.e. −10.7 ‰, i.e. three times smaller than your estimate.

*Overall effect*: Since your estimate $N_2O$ mole fraction enhancement due to emissions (0.15-0.2 nmol mol$^{-1}$) is about twice the $N_2O$ mole fraction depletion due to mixing with stratospheric air (−0.35-0.4 nmol mol$^{-1}$), the net effect is [(−0.4 nmol mol$^{-1}$)(−15 ‰) + (0.2 nmol mol$^{-1}$)(−10.7 ‰)] / (330 nmol mol$^{-1}$) = 0.01 ‰. In other words, the effect due to STE dominates, but is too small to be measured.

**Other corrections:**
Röckmann, not Rockmann

Units and chemical symbols should not be mixed. Please express mass fluxes as "Tg a$^{-1}$ N equivalents" or " Tg a$^{-1}$ N" with a short explanation what this means upon first usage.

ACP requires adhering to the International System of Units (SI) and the Recommendations in the IUPAC Green Book (https://www.atmospheric-chemistry-and-physics.net/for_authors/manuscript_preparation.html)

Please use a suitable (single-letter) quantity symbol for the troposphere-stratosphere air exchange flux (e.g. $F_{TS}$ or $F_{ex}$).

l. 174 & 175: Please use SI units for mole fractions, i.e. nmol mol$^{-1}$ (not ppb).
l. 301: Ditto, but "pmol mol$^{-1}$".

l. 240: What is the remaining 1 %? Even with rounding errors, the values of 78 % and 21 % cannot add up to 100 %.

l. 673: Please replace "species" with "deltas".

Table S2: The uncertainty range for the stratosphere-troposphere exchange rate is too wide. In particular, it cannot be negative.

Tables 2 & S3: You should add the estimates of Röckmann et al. 2003 to the table since you cite our study on various occasions. You may need to make an assumption on the modern tropospheric values on international scales (e.g. use your measurements). An alternative idea would be to express the source signatures of each study relative to the modern value, to avoid biases due to different isotope calibration scales. The source vs. troposphere $\delta$ differences are more robust than the absolute values.

Tables 2 and S3: The internationally accepted abbreviation for "year" is "a" (from Latin annum). You actually use the symbol "a" elsewhere in the manuscript.

Your responses to queries 19 and 20 of referee #2 are missing.

---

## Author Response (AR2)

**Manuscript structure:** To avoid fragmentation and for ease of readability, tables S1 and S2 as well as Figures S3 to S9 should be moved to the main part of the paper. They are key parts of the Results and Discussion sections and in many cases referred to repeatedly in the text. Since Table S3 only adds one additional column with new values compared with Table 2, it should be merged with Table 2.

R: Thanks for the suggestion. We agree that Table S3 should be merged with Table 2. However, for the simplicity and readability of the manuscript, we cannot agree with including Tables S1 and S2 as well as Figures S3 to S9 in the main manuscript. For example, Tables S1 and S2 have been previously published (Harris et al., 2017 for Table S1; Sowers et al., 2002, Röckmann et al., 2003... for Table S2) and have been clearly referenced in our manuscript. For Figures S3 to S9, the contained information was mostly supportive or overlapping with the main manuscript. Therefore, we would rather keep them in the supporting information.

**Data availability:** Please upload all data used for the study to a publicly accessible archive and provide a Digital Object Identifier (DOI). It is not sufficient to have them available on request from the lead author. For details, please refer to https://www.atmospheric-chemistry-andphysics.net/about/data_policy.html

R: OK. This is done in the R2.

**Mixing calculation:** There is an error in your "Estimates of the combined effects from STE and soil emissions on $\delta^{15}N^{SP}$". The effect on $\delta$ due to mixing with stratospheric air is about three times larger than the 5 ‰ value you used. In contrast, the effect due to soil emission is too big and should be about three times smaller. However, overall, the combined effect is too small to be measured.

I think this is an important calculation and you should move it from the supplement to the main text (after correcting it as explained in the following).

*Mixing with stratospheric air*: The effect on tropospheric air can be approximated using the apparent isotopic fractionation $\varepsilon_{app}$ (Kaiser et al. 2006). You can readily derive this from the slope of the Rayleigh fractionation equation at $y = y_T$ (where $y_T$ is the $N_2O$ mole fraction at the point the of mixing between troposphere and stratosphere):

$$\delta = (1+\delta_0)f^{\varepsilon_{app}} - 1 \quad \text{with } f = \frac{y}{y_T}$$

$$\frac{d\delta}{df} = (1+\delta_0)\varepsilon_{app}f^{\varepsilon_{app}-1}$$

$$y_T\frac{d\delta}{dy} = (1+\delta_0)\varepsilon_{app}\left(\frac{y}{y_T}\right)^{\varepsilon_{app}-1}$$

This gives

$$\lim_{y\to y_T}\frac{d\delta}{dy} = \frac{(1+\delta_0)\varepsilon_{app}}{y_T}$$

The derived slope is in agreement with the compact relationship between $\delta$ and $N_2O$ mole fraction plotted in Fig. 5 of Kaiser et al. (2006). For the "site-preference" $\delta^{15}N^{SP}$, $\varepsilon_{app}$ needs to be replaced with the difference between the

$^{15}N/^{14}N$ isotope fractionations at the central and terminal N atoms, $^2\varepsilon_{app}-^1\varepsilon_{app}$ (or $^\alpha\varepsilon_{app}-^\beta\varepsilon_{app}$).

For the lower stratosphere, $\varepsilon_{app}$ ($^{15}N^{SP}$) $\approx -15$ ‰ (Kaiser et al. 2006), i.e. about three times larger than your value of 5 ‰.

*Effect of soil emissions:* I am not sure what you mean by the sentence "The isotopic effect of soil emission (–30‰), which is mainly attributed to the switch from nitrification to denitrification, is taken from Sutka et al. (2006)." It sounds as if you have taken the difference between the $\delta^{15}N^{SP}$ values of denitrification (0 ‰?) and nitrification (+30 ‰?). However, the relevant quantity here is the difference between the $\delta^{15}N^{SP}$ values of soil emissions (+7.3 ‰; Table 3) and tropospheric air (+18 ‰; Fig. 2), i.e. –10.7 ‰, i.e. three times smaller than your estimate. *Overall effect:* Since your estimate $N_2O$ mole fraction enhancement due to emissions (0.15-0.2 nmol mol$^{-1}$) is about twice the $N_2O$ mole fraction depletion due to mixing with stratospheric air (–0.35-0.4 nmol mol$^{-1}$), the net effect is [(–0.4 nmol mol$^{-1}$)(–15 ‰) + (0.2 nmol mol$^{-1}$)(–10.7 ‰)] /(330 nmol mol$^{-1}$) = 0.01 ‰. In other words, the effect due to STE dominates, but is too small to be measured.

R: We really appreciate the editor's explanation and suggestion. Now we have corrected our estimates of the combined effects from STE and soil emissions on $\delta^{15}N^{SP}$ and have implemented details of mixing calculation to M&M. Accordingly, we have rewritten our discussion of seasonal patterns of $N_2O$ isotopic signatures in Section 4.2-R2.

**Other corrections:**

Röckmann, not Rockmann

R: OK, corrected.

Units and chemical symbols should not be mixed. Please express mass fluxes as "Tg a$^{-1}$ N equivalents" or " Tg a$^{-1}$ N" with a short explanation what this means upon first usage.

ACP requires adhering to the International System of Units (SI) and the Recommendations in the IUPAC Green Book (https://www.atmospheric-chemistry-andphysics.net/for_authors/manuscript_preparation.html)

R: OK, we have corrected it as requested.

Please use a suitable (single-letter) quantity symbol for the troposphere-stratosphere air exchange flux (e.g. $F_{TS}$ or $F_{ex}$).

R: OK, we use $F_{ex}$ now.

l. 174 & 175: Please use SI units for mole fractions, i.e. nmol mol$^{-1}$ (not ppb).

l. 301: Ditto, but "pmol mol$^{-1}$".

R: OK, done.

l. 240: What is the remaining 1 %? Even with rounding errors, the values of 78 % and 21 % cannot add up to 100 %.

R: This is a mistake. We have changed it in R2.

l. 673: Please replace "species" with "deltas".

R: This is corrected in R2.

Table S2: The uncertainty range for the stratosphere-troposphere exchange rate is too wide. In particular, it cannot be negative.

R: Sorry for the mistake. It is "(5.37±1.26) x 10$^{17}$" instead and we have corrected this in Table S2.

Tables 2 & S3: You should add the estimates of Röckmann et al. 2003 to the table since you cite our study on various occasions. You may need to make an assumption on the modern tropospheric values on international scales (e.g. use your measurements). An alternative idea would be to express the source signatures of each study relative to the modern value, to avoid biases due to different isotope calibration scales. The source vs. troposphere $\delta$ differences are more robust than the absolute values.

R: OK. We have now included the estimates from Röckmann et al. (2003) for comparison. These $\delta$ values for anthropogenic sources were calculated using the modern tropospheric values from this study. Although expressing the source signatures relative to modern values may help to reduce the biases from different calibration scales, it is rather difficult to justify whether overall mean $\delta$ values observed from the modern troposphere or values from the same years should be chosen for comparison, given that these studies from literature span over two decades. We will consider this approach in future study.

Tables 2 and S3: The internationally accepted abbreviation for "year" is "a" (from Latin annum). You actually use the symbol "a" elsewhere in the manuscript.

R: OK. Replaced.

Your responses to queries 19 and 20 of referee #2 are missing.

19. L 461: Comparing Figure 3(a) in Toyota et al. (2013) with Figure 1a here, it does not look "almost identical", but perhaps comparable. The monthly mixing ratio of $N_2O$ at Jungfraujoch is at maximum in June while in April at Hateruma Island, Japan.
R: Agree. We have changed the description to "comparable".

20. L 464: What are the underlying mechanisms?
R: We actually mean "the explanation of temporal patterns of $N_2O$ isotopic signatures can be complicated by variable $N_2O$ sources". This is now revised in section 4.2.

**List of all relevant changes (referred to Line numbers in R2):**

Line 90: "Tian et al., 2018" to "Tian et al., 2019".

Line 174-175: "ppb" changed to "nmol mol$^{-1}$'.

Line 240: "78% $N_2$ and 21% $O_2$" changed to "78% $N_2$ and 21% $O_2$".

Line 301: "criteria" to "criterion".

Line 302: "ppt" to "pmol mol$^{-1}$".

Line 323-324: "$TS$ex" to "$F$ex".

Line 343-364: Add section 2.8 "Evaluation of the combined effects from STE and soil emission on $\delta^{15}N^{SP}$" for the mixing calculation.

Line 365: "2.8" to "2.9".

Line 425: "Tg $N_2$O-N a$^{-1}$" to "Tg $N_2$O-N a$^{-1}$ equivalents".

Line 497: "almost identical to" to "which is comparable with".

Line 499-500: "complicating the underlying mechanisms for the observed pattern" to "complicating the explanations for the observed temporal pattern".

Line 510: "observe" to "observed".

Line 514: "Figure 5a&b" to "Figure 6a&6b".

Line 521-533: Rewrite the discussion of "minimum of $\delta^{15}N^{SP}$ in late summer and the results from mixing calculation".

Line 642: "Table S3" to "Table 2".

Line 653: Add "Röckmann et al., 2003".

Line 721: "species" to "deltas".

Line 729-730: Add data source "Data for this study have been deposited in a general data repository (https://figshare.com/s/077562ab408dd1bd0880; doi:10.6084/m9.figshare.12032760.v1, 2020)".

Line 845-847: Add reference "Kaiser et al., 2006".

Table 2: Merged with Table S3 and also included the estimates from "Röckmann et al., 2003" for comparison.

[revised manuscript text omitted]
|---|---|---|---|---|---|---|---|---|---|
| Air Sample Origin/age | NH[†] 2014-2018 | NH(FT[η]) 2014-2018 | | FA[‡], IC[†] 1785-1995 | FA NA | FA[‡] 1960-2001 | NH[‡] 1999-2010 | SH[‡], FA[‡] 1940-2005 | FA[‡] 1940-2008 |
| $\alpha$-* | 0.0154±0.004 | 0.0154±0.004 | 0.65 nmol mol$^{-1}$ | 0.0111 to 0.0128 | NA | NA | NA | NA | NA |
| $F_{anth,2018}$ (Tg N ya$^{-1}$) | 8.6±0.6 | 8.5±0.6 | NA[‡] | 4.2 to 5.7 | 6.9 | NA | 5.5 | 6.6 | 5.4±1.7 |
| $\delta^{15}N^{bulk}$-anth (‰) | -8.6±4 | -8.5±4 | 0.23 | -7 to -13 | -11.4 | -11.6 | -9.84 | -15.6±1.2 | -18.2±2.6 |
| $\delta^{18}O$-anth (‰) | 34.8±3 | 34.3±3 | 0.22 | 17 to 26 | 31.7 | NA | 35.95 | 32.0±1.3 | 27.2±2.6 |
| $\delta^{15}N^{SP}$-anth (‰) | 10.7±4 | 10.7±4 | 0.50 | NA | 11.3 | NA | 8.52 | 13.1±9.4 | 18.0±8.6 |

[revised manuscript text omitted]

---

## Author Response (AR3)

Dear Editor,

We thank you again for reviewing our manuscript and your valuable suggestions to improve our manuscript. We also appreciate your opinions on how to use units and symbols. Here, you will find our revised manuscript with a final version and a version with track-change. As you requested, we have revised the typo and have changed the structure. In this round, we accept most suggestions from you except for Fig. S7 (R2). Given that Fig. S7 is only used once in the discussion and was derived from Personal Communication (Henne et al.), we prefer to keep them in the supplementary.

We are happy that this manuscript will finally be accepted for publication. In future, our group will for sure submitted more manuscripts to ACP.

Longfei Yu
On behalf of all co-authors

[revised manuscript text omitted]